# PRMT5 regulates ovarian follicle development by facilitating *Wt1* translation

**Min Chen[1,2,3,4†‡], Fangfang Dong[1,2,4†], Min Chen[5‡], Zhiming Shen[1,2,4], Haowei Wu[1,2,4], Changhuo Cen[1,2,4], Xiuhong Cui[1,2], Shilai Bao[6], Fei Gao[1,2,3,4]\***

[1]State Key Laboratory of Stem Cell and Reproductive Biology, Institute of Zoology, Chinese Academy of Sciences, Beijing, China; [2]Institute for Stem Cell and Regeneration, Chinese Academy of Sciences, Beijing, China; [3]Beijing Institute for Stem Cell and Regenerative Medicine, Beijing, China; [4]University of Chinese Academy of Sciences, Beijing, China; [5]Guangdong and Shenzhen Key Laboratory of Male Reproductive Medicine and Genetics, Institute of Urology, Peking University Shenzhen Hospital, Shenzhen, China; [6]State Key Laboratory of Molecular Developmental Biology, Institute of Genetics and Developmental Biology, Chinese Academy of Sciences, Beijing, China

**Abstract** Protein arginine methyltransferase 5 (*Prmt5*) is the major type II enzyme responsible for symmetric dimethylation of arginine. Here, we found that PRMT5 was expressed at high level in ovarian granulosa cells of growing follicles. Inactivation of *Prmt5* in granulosa cells resulted in aberrant follicle development and female infertility. In *Prmt5*-knockout mice, follicle development was arrested with disorganized granulosa cells in which WT1 expression was dramatically reduced and the expression of steroidogenesis-related genes was significantly increased. The premature differentiated granulosa cells were detached from oocytes and follicle structure was disrupted. Mechanism studies revealed that *Wt1* expression was regulated by PRMT5 at the protein level. PRMT5 facilitated IRES-dependent translation of *Wt1* mRNA by methylating HnRNPA1. Moreover, the upregulation of steroidogenic genes in *Prmt5*-deficient granulosa cells was repressed by *Wt1* overexpression. These results demonstrate that PRMT5 participates in granulosa cell lineage maintenance by inducing *Wt1* expression. Our study uncovers a new role of post-translational arginine methylation in granulosa cell differentiation and follicle development.

**\*For correspondence:**
gaof@ioz.ac.cn

†These authors contributed equally to this work

‡Two of the authors have the name 'Min Chen'

**Competing interest:** The authors declare that no competing interests exist.

## Introduction

Follicles are the basic functional units in the ovaries. Each follicle consists of an oocyte, the surrounding granulosa cells, and theca cells in the mesenchyme. The interaction between oocytes and somatic cells is crucial for follicle development. Follicle maturation experiences primordial, primary, secondary, and antral follicular stages. Primordial follicles are formed shortly after birth via breakdown of oocyte syncytia. Each primordial follicle is composed of an oocyte surrounded by a single layer of flattened pregranulosa cells that remains in a dormant phase until being recruited into the primary stage under the influence of two main signaling pathways (*Monniaux, 2016*). Once activated, flattened granulosa cells become cuboidal, and follicles continue to grow through proliferation of granulosa cells and enlargement of oocytes. Development of high-quality oocytes is important for female reproductive health and fertility (*Jagarlamudi and Rajkovic, 2012*; *Liu et al., 2010*; *Monniaux, 2016*; *Richards and Pangas, 2010*). Although gonadotropin, follicle-stimulating hormone (FSH), and luteinizing hormone (LH) are important for the growth of antral follicles, the early stages of follicle development are driven

**eLife digest** Infertility in women can be caused by many factors, such as defects in the ovaries. An important part of the ovaries for fertility are internal structures called follicles, which house early forms of egg cells. A follicle grows and develops until the egg is finally released from the ovary into the fallopian tube, where the egg can then be fertilised. In the follicle, an egg is surrounded by other types of cells, such as granulosa cells. The egg and neighbouring cells must maintain healthy contacts with each other, otherwise the follicle can stop growing and developing, potentially causing infertility.

The development of a follicle depends on an array of proteins. For example, the transcription factor WT1 controls protein levels by activating other genes and their proteins and is produced in high numbers by granulosa cells at the beginning of follicle development. Although WT1 levels dip towards the later stages of follicle development, insufficient levels can lead to defects. So far, it has been unclear how levels of WT1 in granulose cells are regulated.

Chen, Dong et al. studied mouse follicles to reveal more about the role of WT1 in follicle development. The researchers measured protein levels in mouse granulosa cells as the follicles developed, and discovered elevated levels of PRMT5, a protein needed for egg cells to form and survive in the follicles. Blocking granulosa cells from producing PRMT5 led to abnormal follicles and infertility in mice. Moreover, mice that had been engineered to lack PRMT5 developed abnormal follicles, where the egg and surrounding granulosa cells were not attached to each other, and the granulosa cells had low levels of WT1. Further experiments revealed that PRMT5 controlled WT1 levels by adding small molecules called methyl groups to another regulatory protein called HnRNPA1.

The addition of methyl groups to genes or their proteins is an important modification that takes place in many processes within a cell. Chen, Dong et al. reveal that this activity also plays a key role in maintaining healthy follicle development in mice, and that PRMT5 is necessary for controlling WT1. Identifying all of the intricate mechanism involved in regulating follicle development is important for finding ways to combat infertility.

by a local oocyte-granulosa cell dialog. Abnormalities in this process may lead to follicle growth arrest or atresia (*Oktem and Urman, 2010*; *Richards and Pangas, 2010*).

Granulosa cells are derived from progenitors of the coelomic epithelium that direct sexual differentiation at the embryonic stage and support oocyte development postnatally (*Liu et al., 2010*; *Smith et al., 2014*). Theca-interstitial cell differentiation occurs postnatally along with the formation of secondary follicles. The steroid hormone produced by theca-interstitial cells plays important roles in follicle development and maintenance of secondary sexual characteristics (*Liu et al., 2010*). The Wilms' tumor (WT) suppressor gene *Wt1* is a nuclear transcription factor indispensable for normal development of several tissues. In gonads, *Wt1* is mainly expressed in ovarian granulosa cells and testicular Sertoli cells. During follicle development, *Wt1* is expressed at high levels in granulosa cells of primordial, primary, and secondary follicles, but its expression is decreased in antral follicles, suggesting that it might be a repressor of ovarian differentiation genes in the granulosa cells (*Hsu et al., 1995*). Our previous studies demonstrated that *Wt1* is required for the lineage specification and maintenance of Sertoli and granulosa cells (*Cen et al., 2020*; *Chen et al., 2017*). However, the underlying mechanism that regulates the expression of *Wt1* in granulosa cells is unknown.

Protein arginine methyltransferase 5 (PRMT5) is a member of the PRMT family that catalyzes the transfer of methyl groups from S-adenosylmethionine to a variety of substrates and is involved in many cellular processes, such as cell growth, differentiation, and development (*Di Lorenzo and Bedford, 2011*; *Karkhanis et al., 2011*; *Stopa et al., 2015*). PRMT5 is the predominant type II methyltransferase that catalyzes the formation of most symmetric dimethylarginines (SDMAs) in the cells and regulates gene expression at the transcriptional and posttranscriptional levels (*Karkhanis et al., 2011*). PRMT5 forms a complex with its substrate-binding partner, the WD-repeat protein MEP50 (or WDR77), which greatly enhances the methyltransferase activity of PRMT5 by increasing its affinity for protein substrates (*Stopa et al., 2015*).

In gonad development, inactivation of *Prmt5* specifically in primordial germ cells (PGCs) causes massive loss of PGCs (*Kim et al., 2014*; *Li et al., 2015*; *Wang et al., 2015*). PRMT5 promotes PGC survival by regulating RNA splicing (*Li et al., 2015*) and suppressing transposable elements at the time

of global DNA demethylation (*Kim et al., 2014*). In this study, we found that PRMT5 is expressed at high level in ovarian granulosa cells of growing follicles and the expression level changes with follicle development, suggesting that PRMT5 in granulosa cells plays a role in follicle development. To test the function of PRMT5 in granulosa cells, we specifically inactivated *Prmt5* in granulosa cells using *Sf1*<sup>+/cre</sup>. The *Sf1*<sup>+/cre</sup> mouse expresses *Cre* recombinase in the adrenogonadal primordium at 10 dpc, the precursors for cortical cells in the adrenals and somatic cells in the gonads (*Bingham et al., 2006*; *Huang and Yao, 2010*). We found that *Prmt5*<sup>flox/flox</sup>;*Sf1*<sup>+/cre</sup> female mice were infertile and that follicles were arrested at the secondary stage. The expression of WT1 was dramatically reduced, and the granulosa cells in secondary follicles began to express steroidogenic genes. Further studies revealed that PRMT5 regulates follicle development by facilitating *Wt1* translation.

## Results

### Deletion of *Prmt5* in granulosa cells caused aberrant ovary development and female infertility

The expression of PRMT5 in ovarian granulosa cells was examined by immunofluorescence. As shown in *Figure 1—figure supplement 1*, PRMT5 (red) was expressed in oocytes, but no PRMT5 signal was detected in the granulosa cells of primordial follicles (A, A', white arrows). PRMT5 started to be expressed in granulosa cells of primary follicles (B, B', white arrows) and was continuously expressed in granulosa cells of secondary follicles (C, C', white arrows), and antral follicles (D, D', white arrows), but its expression decreased significantly in the corpus luteum (E, E', white arrows). To test the functions of PRMT5 in granulosa cell development, we specifically deleted *Prmt5* in granulosa cells by crossing *Prmt5*<sup>flox/flox</sup> mice with *Sf1*<sup>+/cre</sup> transgenic mice. In *Prmt5*<sup>flox/flox</sup>;*Sf1*<sup>+/cre</sup> female mice, PRMT5 expression was completely absent from granulosa cells (*Figure 1—figure supplement 2B,D*, arrows), whereas the expression of PRMT5 in oocytes and interstitial cells was not affected, suggesting that *Prmt5* was specifically deleted in granulosa cells.

No obvious developmental abnormalities were observed in adult *Prmt5*<sup>flox/flox</sup>;*Sf1*<sup>+/cre</sup> mice (*Figure 1A*). However, the female mice were infertile with atrophic ovaries (*Figure 1B*, *Figure 1—figure supplement 3*). The results of immunohistochemistry showed growing follicles at different stages in control ovaries at 2 months of age (*Figure 1C*). In contrast, only a small number of follicles and few corpora lutea were observed in *Prmt5*<sup>flox/flox</sup>;*Sf1*<sup>+/cre</sup> mice (*Figure 1D*). We further examined follicle development in *Prmt5*<sup>flox/flox</sup>;*Sf1*<sup>+/cre</sup> mice at different developmental stages. As shown in *Figure 1*, a large number of growing follicles at the primary and secondary stages were observed in *Prmt5*<sup>flox/flox</sup>;*Sf1*<sup>+/cre</sup> mice (F) at 2 weeks, which was comparable to the situation in control ovaries (E). Many follicles progressed to antral follicle stages in control mice at 3 weeks (G), whereas the development of follicles in *Prmt5*<sup>flox/flox</sup>;*Sf1*<sup>+/cre</sup> mice was arrested at the secondary stage. Almost no antral follicle was observed (*Figure 1—figure supplement 4B*) and aberrant granulosa cells were evident (*Figure 1H*). The defects in follicle development were more obvious at 4 and 5 weeks (*Figure 1J and L*, *Figure 1—figure supplement 4C and D*). The number of granulosa cells around oocytes was dramatically reduced, and follicle structure was disrupted.

### The identity of granulosa cells in *Prmt5*<sup>flox/flox</sup>;*Sf1*<sup>+/cre</sup> mice was changed

To explore the underlying mechanism that caused the defects in follicle development in *Prmt5*<sup>flox/flox</sup>;*Sf1*<sup>+/cre</sup> mice, the expression of granulosa cell-specific genes was analyzed by immunohistochemistry. As shown in *Figure 2*, FOXL2 protein was expressed in the granulosa cells of both control (A, C) and *Prmt5*<sup>flox/flox</sup>;*Sf1*<sup>+/cre</sup> mice (B, D) at P14 and P18. WT1 protein was expressed in granulosa cells of primordial, primary, and secondary follicles in control mice at P14 and P18 (E, E', G, G', arrows). WT1 was also detected in the follicles of *Prmt5*<sup>flox/flox</sup>;*Sf1*<sup>+/cre</sup> mice at P14 (F, arrow). However, not all the granulosa cells were WT1-positive; some of them were WT1-negative (F', arrowheads). The WT1 signal was almost completely absent from the majority of granulosa cells in *Prmt5*<sup>flox/flox</sup>;*Sf1*<sup>+/cre</sup> mice at P18 (H, H'); very few granulosa cells were WT1-positive (H', arrowheads). We also found that the granulosa cells in control ovaries were cuboidal and well-organized (G', arrow). In contrast, the granulosa cells in *Prmt5*<sup>flox/flox</sup>;*Sf1*<sup>+/cre</sup> mice were flattened (H', dashed line circle) and were indistinguishable from surrounding stromal cells. The decreased WT1 protein expression in *Prmt5*-deficient granulosa cells was also confirmed by FOXL2/WT1 double staining (*Figure 2—figure supplement 1*).

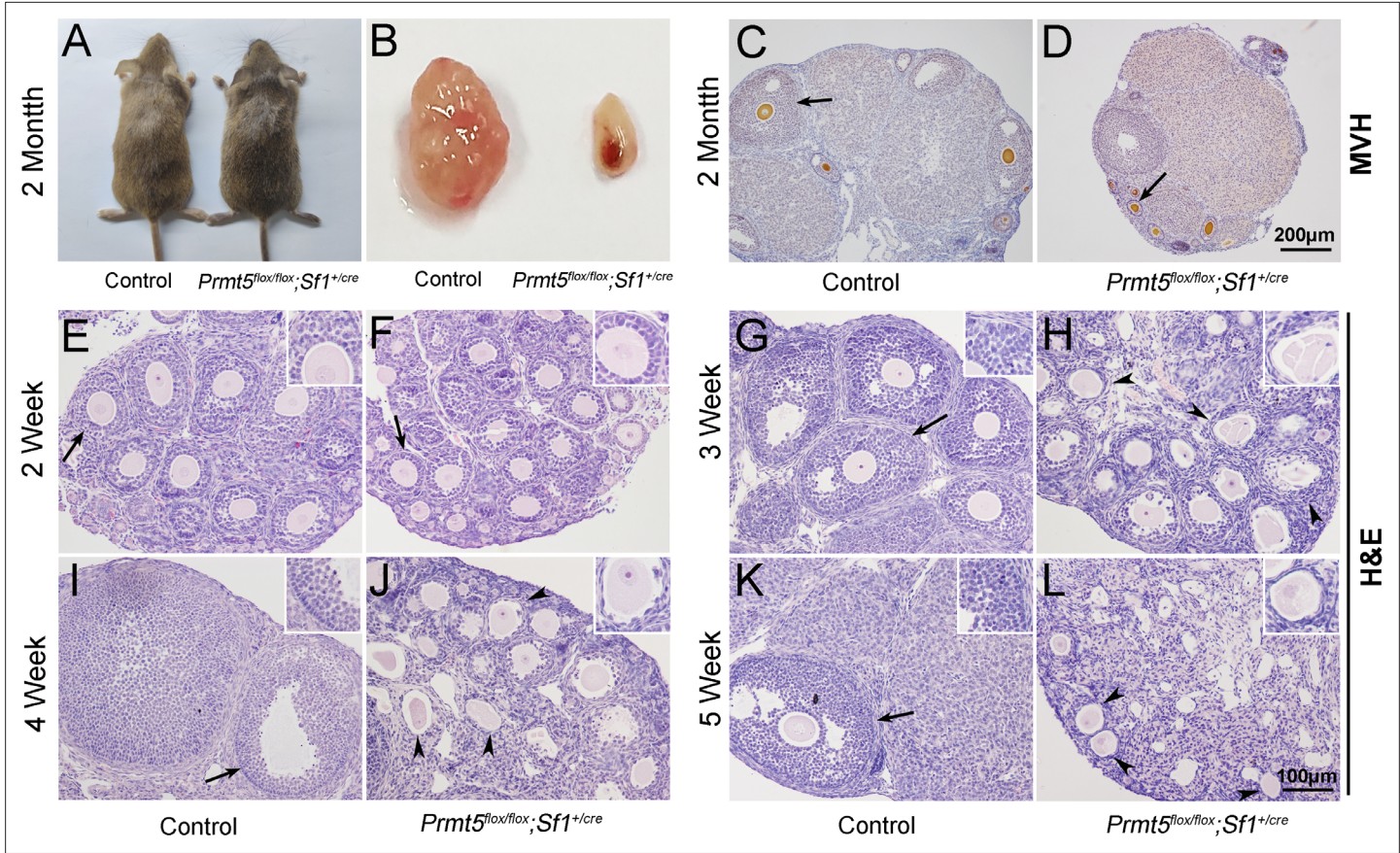

**Figure 1.** Loss of *Prmt5* in granulosa cells caused aberrant follicle development and female infertility. No developmental abnormalities were observed in *Prmt5^{flox/flox}*;*Sf1^{+/cre}* mice (**A**) at 2 months of age, and the ovary size was dramatically reduced (**B**). Histology of ovaries from control (**C**) and *Prmt5^{flox/flox}*;*Sf1^{+/cre}* mice (**D**) at 2 months of age. The histology of ovarian follicles was grossly normal in *Prmt5^{flox/flox}* mice at 2 weeks (**F**, black arrows). Defects in follicle development were observed in *Prmt5*-mutant mice at 3 weeks (**H**, black arrowheads). Aberrant ovarian follicles with disorganized granulosa cells were observed in *Prmt5^{flox/flox}*;*Sf1^{+/cre}* mice at 4 (**J**, black arrowheads) and 5 (**L**, black arrowheads) weeks of age. (**E**), (**G**), (**I**), (**K**) are the histology of ovarian follicles in control mice respectively at 2, 3, 4, and 5 weeks.

The online version of this article includes the following source data and figure supplement(s) for figure 1:

**Figure supplement 1.** PRMT5 was expressed in granulosa cells of growing follicles.

**Figure supplement 1—source data 1.** Source data for *Figure 1—figure supplements 3 and 4*.

**Figure supplement 2.** *Prmt5* was deleted in granulosa cells of *Prmt5^{flox/flox}*;*Sf1^{+/cre}* mice.

**Figure supplement 3.** Female *Prmt5^{flox/flox}*;*Sf1^{+/cre}* mice were infertile.

**Figure supplement 4.** Follicle development was arrested at the secondary stage.

*Wt1* plays a critical role in granulosa cell development, and mutation of *Wt1* leads to pregranulosa cell-to-steroidogenic cell transformation (*Cen et al., 2020*; *Chen et al., 2017*). Therefore, we further examined the expression of steroidogenic genes in *Prmt5*-deficient granulosa cells at P18. As shown in *Figure 3*, in control ovaries, 3β-HSD (3β-hydroxysteroid dehydrogenase, also known as Hsd3B1) and CYP11A1 (cytochrome P450, family 11, subfamily a, polypeptide 1, also known as P450scc) were expressed in theca-interstitial cells (A, C, arrowheads). In addition to theca-interstitial cells, 3β-HSD (B, green, arrows) and CYP11A1 (D, red, arrows) were also detected in the granulosa cells of *Prmt5^{flox/flox}*;*Sf1^{+/cre}* mice. We also examined the expression of SF1 (steroidogenic factor 1, also known as NR5A1), which is a key regulator of steroid hormone biosynthesis (*Ikeda et al., 1993*). As expected, SF1 was expressed only in theca-interstitial cells of control ovaries (E, red, arrowheads), whereas a high level of SF1 expression was detected in *Prmt5*-deficient granulosa cells (F, red, arrows), suggesting that the identity of granulosa cells was changed. The follicle structure was destroyed as indicated by disorganized Laminin staining (H, arrows). The proliferation and apoptosis of *Prmt5*-deficient granulosa cells

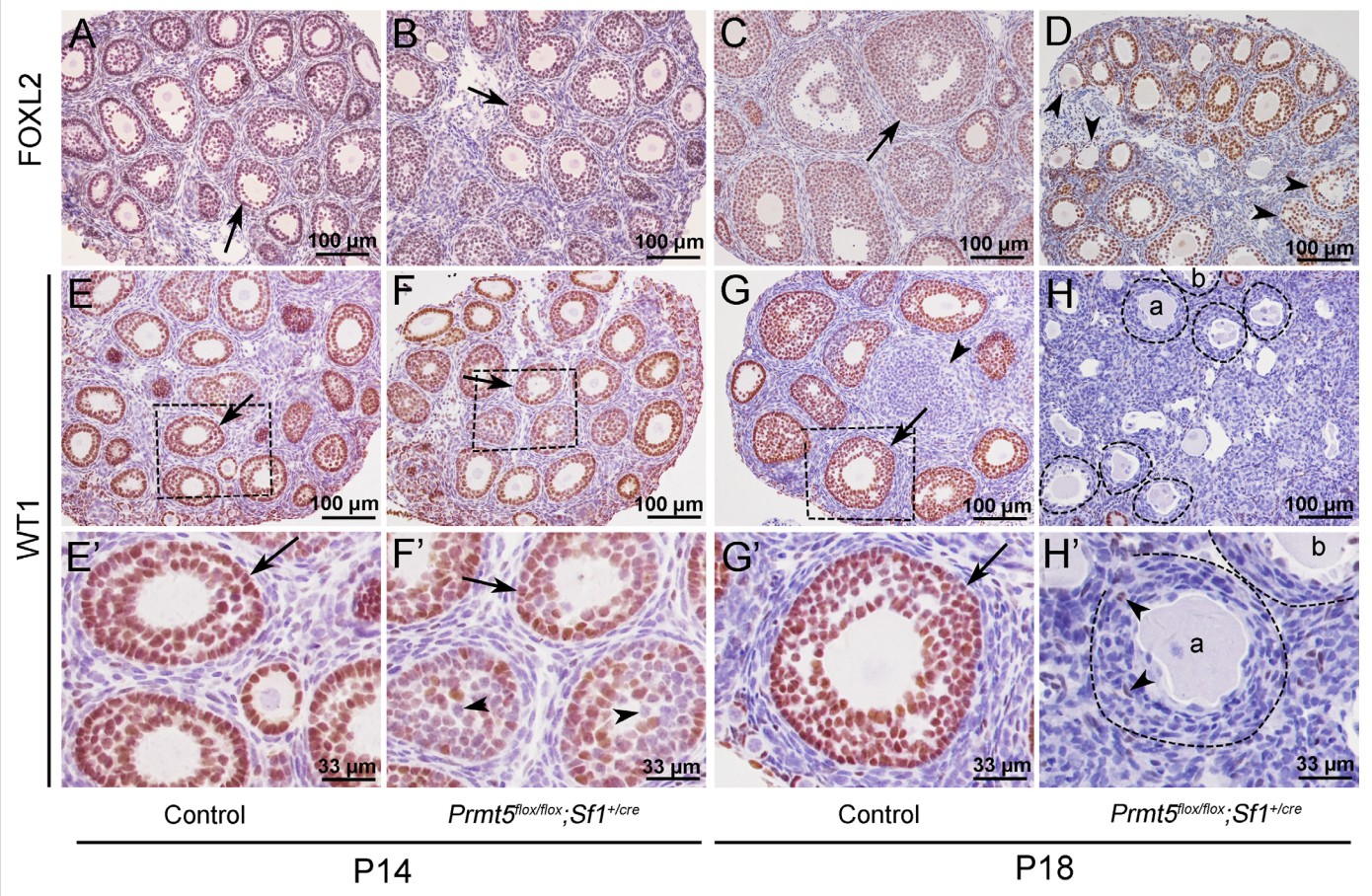

**Figure 2.** The expression of WT1 was dramatically reduced in the granulosa cells of *Prmt5flox/flox;Sf1+/cre* mice at P18. The expression of FOXL2 and WT1 in granulosa cells of control and *Prmt5flox/flox;Sf1+/cre* mice was examined by immunohistochemistry. FOXL2 protein was expressed in the granulosa cells of both control (**A, C**) and *Prmt5flox/flox;Sf1+/cre* mice (**B, D**) at P14 and P18. WT1 protein was expressed in granulosa cells of primordial, primary, and secondary follicles in control mice at P14 and P18 (**E, E', G, G'**, black arrows). WT1 expression was absent from most granulosa cells in *Prmt5flox/flox;Sf1+/cre* mice at P18 (**H, H'**); only very few granulosa cells were WT1-positive (**H'**, black arrowheads). (**E'–H'**) are the magnified views of (**E–H**) respectively.

The online version of this article includes the following figure supplement(s) for figure 2:

**Figure supplement 1.** WT1 expression was decreased significantly in *Prmt5flox/flox;Sf1+/cre* granulosa cells at P18.

were analyzed by BrdU incorporation and TUNEL assays. As shown in *Figure 3—figure supplement 1*, the numbers of TUNEL-positive cells and BrdU-positive granulosa cells were not changed in *Prmt-5flox/flox;Sf1+/cre* ovaries compared to control ovaries at P14 and P18.

To further confirm the above results, follicles were dissected from the ovaries of 2-week-old mice and cultured in vitro. As shown in *Figure 3—figure supplement 2*, the histology of follicles from *Prmt5flox/flox;Sf1+/cre* mice was comparable to that of control follicles at D2. Proliferation of granulosa cells in control follicles was observed at D4, and the follicles developed to the preovulatory stage with multiple layers of granulosa cells after 9 days of culture (A–C, G, H). The granulosa cells were detached from oocytes in *Prmt5flox/flox;Sf1+/cre* follicles at D4 (E, and a magnified view in L), and no colonized granulosa cells were observed after 9 days of culture (D–F, J, K). Most of the granulosa cells were attached to the culture dishes just like the interstitial cells, and denuded oocytes were observed after 3 days of culture (E, F, J, K, and a magnified view in L).

To further verify the differential expression of granulosa cell-specific and steroidogenic genes in *Prmt5*-deficient granulosa cells, granulosa cells were isolated at P18, and gene expression was analyzed by western blot and real-time PCR analyses. As shown in *Figure 4A,B*, the protein levels of PRMT5 and its interacting partner MEP50 were decreased dramatically in *Prmt5flox/flox;Sf1+/cre* granulosa cells, as expected. The protein level of WT1 was significantly reduced in *Prmt5*-deficient granulosa cells. Surprisingly, the mRNA level of *Wt1* was not changed in *Prmt5*-deficient granulosa cells (C).

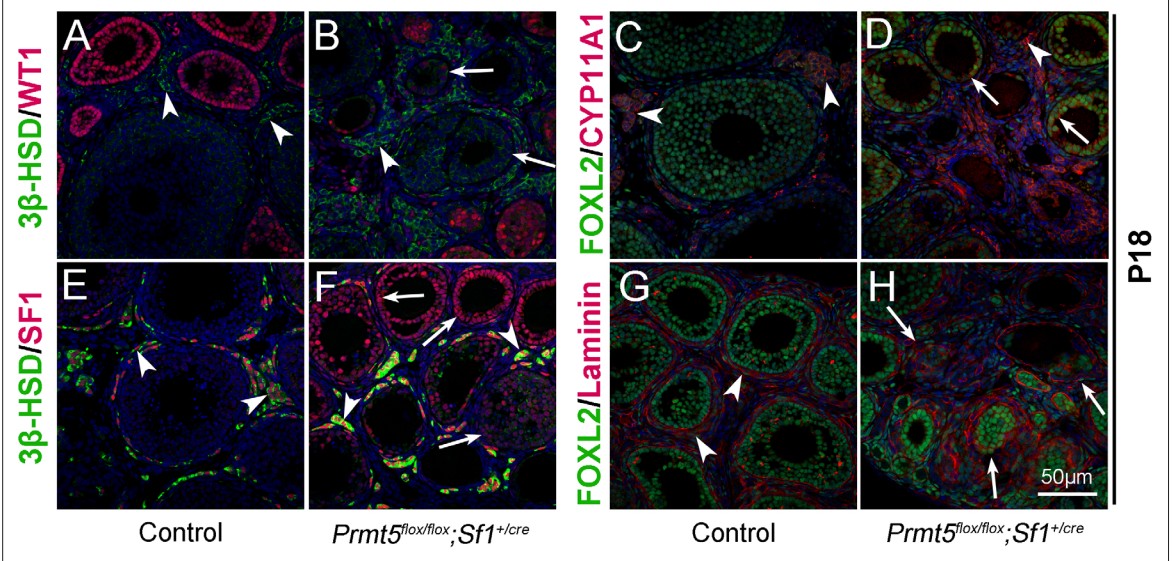

**Figure 3.** The identity of granulosa cells in *Prmt5^flox/flox^;Sf1^+/cre^* mice was changed. The expression of 3β-HSD, WT1, FOXL2, CYP11A1, and SF1 in ovaries of control and *Prmt5^flox/flox^;Sf1^+/cre^* mice at P18 was examined by immunofluorescence. In control ovaries, 3β-HSD (**A**), CYP11A1 (**C**), and SF1 (**E**) were expressed only in theca-interstitial cells (white arrowheads). In the ovaries of *Prmt5^flox/flox^;Sf1^+/cre^* mice, 3β-HSD (**B**), CYP11A1 (**D**), and SF1 (**F**) were also detected in granulosa cells (white arrows). Compared to the intact follicle structure in control ovaries (arrowheads, **G**), the follicle structure was disordered in *Prmt5^flox/flox^;Sf1^+/cre^* ovaries (arrows, **H**) as shown by Laminin expression. DAPI (blue) was used to stain the nuclei.

The online version of this article includes the following figure supplement(s) for figure 3:

**Figure supplement 1.** The apoptosis and proliferation of granulosa cells were not changed in *Prmt5^flox/flox^;Sf1^+/cre^* mice at P14 and P18.

**Figure supplement 2.** Aberrant development of in vitro-cultured *Prmt5^flox/flox^;Sf1^+/cre^* follicles.

FOXL2 expression was also decreased, but the difference was not significant. The expression of the steroidogenic genes CYP11A1, StAR, and NR5A1 was significantly increased in *Prmt5*-deficient granulosa cells, consistent with the immunostaining results. The mRNA levels of these genes were also significantly increased (C). *Cyp19a1* is expressed in granulosa cells and its expression is increased in antral and preovulatory follicles (**Doody et al., 1990**; **Stocco, 2008**). We found that the mRNA level of *Cyp19a1* was also significantly increased in *Prmt5*-deficient granulosa cells (C). We also examined the functions of PRMT5 by treating granulosa cells with the PRMT5-specific inhibitor EPZ015666. The protein level of WT1 was significantly reduced after EZP015666 treatment, whereas the mRNA level was not changed. The expression of steroidogenic genes was significantly increased at both the protein and mRNA levels after EZP015666 treatment (**Figure 4D, E and F**). These results were consistent with those in *Prmt5*-deficient granulosa cells, indicating that the effect of PRMT5 on granulosa cells was dependent on its methyltransferase activity. These results suggest that PRMT5 is required for maintenance of granulosa cell identity and that inactivation of this gene causes premature luteinization of granulosa cells.

## The expression of WT1 was regulated by PRMT5 at the translational level

PRMT5 has been reported to regulate the translation of several genes in an IRES-dependent manner (**Gao et al., 2017**; **Holmes et al., 2019**). Internal ribosome entry sites (IRESs) are secondary structures in the 5′UTR that directly recruit the ribosome cap independently and initiate translation without cap binding and ribosome scanning (**Baird et al., 2006**; **Coldwell et al., 2000**; **Stoneley and Willis, 2004**). *Wt1* 5′UTR is 268 bp, GC-rich (68%) and contains seven CUG codons and one AUG codon. These features usually act as strong barriers for ribosome scanning and conventional translation initiation. Translation initiation in a number of these mRNAs is achieved via IRES-mediated mechanisms (**Stoneley and Willis, 2004**). To test whether the *Wt1* 5′UTR contains an IRES element, we utilized a pRF dicistronic reporter construct in which upstream Renilla luciferase is translated cap-dependently, whereas downstream Firefly luciferase is not translated unless a functional IRES is present. A stable

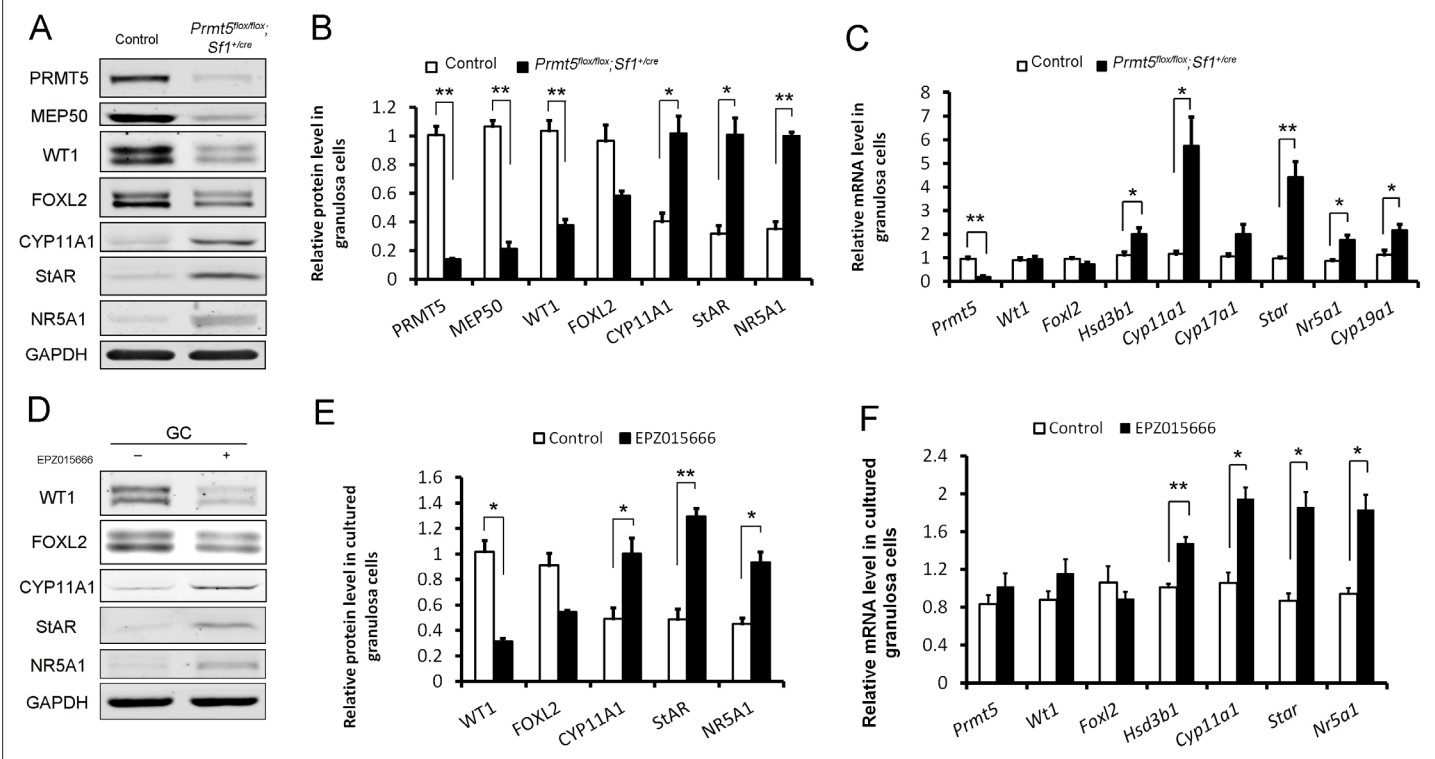

**Figure 4.** Differentially expressed genes in *Prmt5*-deficient granulosa cells. Western blot (**A, B**) and real-time PCR analyses (**C**) of the indicated genes in granulosa cells isolated from control or *Prmt5^{flox/flox};Sf1^{+/cre}* ovaries at P18. Western blot (**D, E**) and real-time PCR analyses (**F**) of the indicated genes in granulosa cells treated with DMSO or EPZ015666 (5 µM) for 5 days. The protein expression of three independent experiments in western blot analysis was quantified and normalized to that of GAPDH (**B, E**). The data are presented as the mean ± SEM. For (**B, E, F**), n = 3; for (**C**), n = 5. *p<0.05. **p<0.01.

The online version of this article includes the following figure supplement(s) for figure 4:

**Source data 1.** Source data for *Figure 4B, C, E and F*.

hairpin structure upstream of Renilla luciferase minimizes cap-dependent translation (***Coldwell et al., 2000***, ***Figure 5A***). *Wt1* 5′UTR was inserted into the intercistronic region between Renilla and Firefly luciferase (named pRWT1F), and primary granulosa cells were transfected with pRF or pRWT1F. The Firefly/Renilla luciferase activity ratio was analyzed 24 hr later. As shown in ***Figure 5B***, the Firefly/Renilla luciferase activity ratio was dramatically increased in pRWT1F-transfected cells compared to pRF-transfected cells. In contrast, the Firefly/Renilla luciferase activity ratio was not increased when *Wt1* 5′UTR was inserted in the reverse direction (pRWT1-RevF; ***Figure 5B***). The luciferase activity was dramatically increased with insertion of *Ccnd1* 5′UTR as a positive control, which has been reported to contain an IRES element in the 5′UTR (***Shi et al., 2005***). These results suggest that *Wt1* 5′UTR probably contains an IRES element.

To verify that Firefly luciferase protein was synthesized by translation of an intact dicistronic transcript instead of a monocistronic mRNA generated by cryptic splicing or promoter within the dicistronic gene (***Kunze et al., 2016***), mRNA from pRF- or pRWT1F-transfected cells was treated with DNase, reverse-transcribed, and then amplified with primers binding to the 5′ end of Renilla luciferase and 3′ end of Firefly luciferase open reading frame spanning the whole transcript. Only one band was detected in both cells with the expected molecular weight (***Figure 5—figure supplement 1A***). Moreover, qPCR analysis of Firefly and Renilla luciferase mRNA levels also showed that the Firefly/Renilla luciferase mRNA ratio was not different between pRF- and pRWT1F-transfected cells (***Figure 5— figure supplement 1B***), further excluding the possibility that insertion of the *Wt1* 5′UTR into pRF generated a monocistronic Firefly ORF.

To examine which part of *Wt1* 5′UTR contributes to its IRES activity, *Wt1* 5′UTR was divided into three fragments and respectively inserted into pRF construct (pRWT1F −268 to −158, pRWT1F −198 to

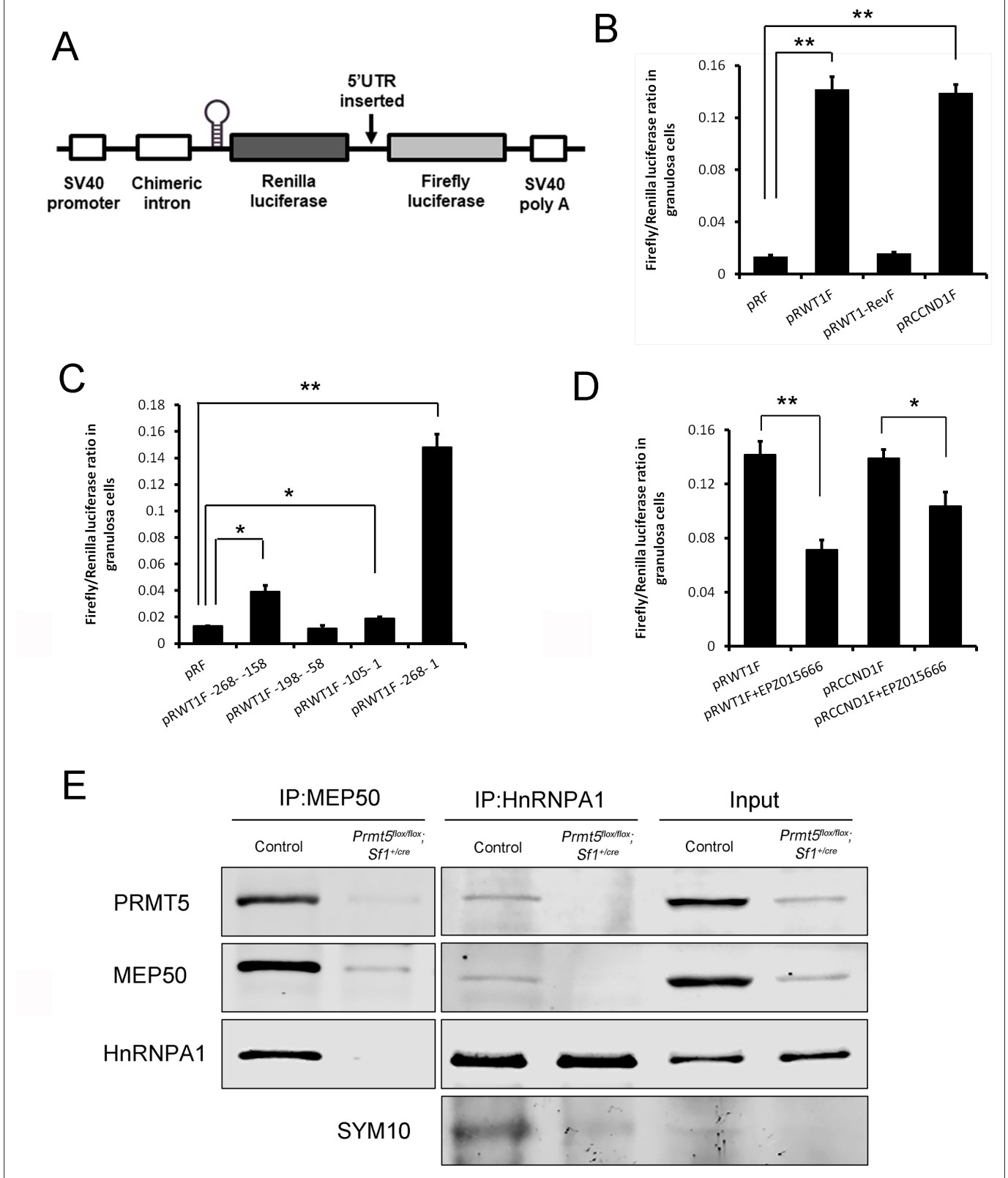

**Figure 5.** PRMT5 regulated translation of *Wt1* mRNA by inducing internal ribosome entry site (IRES) activity in the 5'UTR. (**A**) Schematic representation of the dicistronic reporter construct. (**B**) *Wt1* 5'UTR has IRES activity. Cultured primary granulosa cells were transfected with pRF, pRWT1F (pRF with the *Wt1* 5'UTR inserted), pRWT1-RevF (pRF with the *Wt1* 5'UTR inserted in reverse orientation), or pRCCND1F (pRF with the *Ccnd1* 5'UTR inserted). The Firefly and Renilla luciferase activities were measured 24 hr later, and the ratios of Firefly luciferase activity to Renilla luciferase activity were calculated.

*Figure 5 continued on next page*

*Figure 5 continued*

(**C**) The full length of *Wt1* 5'UTR is required for maximal luciferase activity. Three fragments of *Wt1* 5'UTR were inserted into pRF construct (pRWT1F –268 to –158, pRWT1F –198 –to –58, pRWT1F –105 to –1) and the constructs were transfected into primary granulosa cells. 24 hr later, the cells were harvested for luciferase activity analysis. (**D**) Luciferase activity was decreased in primary granulosa cells treated with the PRMT5 inhibitor EPZ015666. Isolated granulosa cells were treated with DMSO or EPZ015666 for 4 days. The day granulosa cells were isolated was denoted as day 1. On day 4, granulosa cells were transfected with pRWT1F or pRCCND1F. 24 hr later, the cells were harvested for luciferase activity analysis. The ratios of Firefly luciferase activity to Renilla luciferase activity were calculated. In (**B–D**), the data are presented as the mean ± SEM, n = 4. *p<0.05. **p<0.01. (**E**) Coimmunoprecipitation experiments were conducted in control and *Prmt5^flox/flox*;*Sf1^+/cre* granulosa cells. In control granulosa cells, HnRNPA1 was pulled down with an antibody against the PRMT5-associated protein MEP50; PRMT5 and MEP50 were pulled down by an HnRNPA1 antibody. The symmetric dimethylation of HnRNPA1 was significantly decreased in *Prmt5^flox/flox*;*Sf1^+/cre* granulosa cells. Blots are representative of three independent experiments.

The online version of this article includes the following figure supplement(s) for figure 5:

**Source data 1.** Source data for *Figure 5B, C and D*.

**Figure supplement 1.** The increased luciferase activity of pRWT1F was not due to a monocistronic Firefly open reading frame generated by cryptic splicing or promoter within the dicistronic gene.

**Figure supplement 1—source data 1.** Source data for *Figure 5—figure supplement 1B and C*.

---

–58, pRWT1F –105 to 1). These constructs were transfected into primary granulosa cells (*Figure 5C*). No IRES activity was detected for middle fragment (–198 to –58). Although the luciferase activity of pRWT1F –268 to –158 and pRWT1F –105 to –1 was significantly increased compared with control pRF, they were much lower than that of the full-length 5'UTR, suggesting that the full length of *Wt1* 5~UTR is required for maximal IRES activity.

To investigate the effect of PRMT5 on *Wt1* IRES activity, granulosa cells were treated with EPZ015666 for 4 days, we found that *Wt1* IRES activity was decreased significantly after EPZ015666 treatment (*Figure 5D*). As a positive control, the IRES activity of *Ccnd1* 5'UTR was also significantly decreased after EPZ015666 treatment, which was consistent with previous study (*Gao et al., 2017*, *Figure 5D*). We also checked *Wt1* IRES activity in *Prmt5^flox/flox*;*Sf1^+/cre* granulosa cells. As expected, *Wt1* IRES activity was significantly decreased in *Prmt5*-deficient granulosa cells compared with control granulosa cells (*Figure 5—figure supplement 1C*). These results indicate that PRMT5 regulates *Wt1* expression at the translational level by inducing its IRES activity in granulosa cells.

## Wt1 IRES activity was regulated by PRMT5 through methylation of HnRNPA1

IRES-mediated translation depends on IRES *trans*-acting factors (ITAFs), which function by associating with the IRES and either facilitate the assembly of initiation complexes or alter the structure of the IRES (*Jo et al., 2008*; *Kunze et al., 2016*). Heterogeneous nuclear ribonucleoprotein A1 (HnRNPA1) is a well-studied RNA binding protein that plays important roles in pre-mRNA and mRNA metabolism (*Dreyfuss et al., 2002*). HnRNPA1 is also an ITAF that has been reported to regulate the IRES-dependent translation of many genes, such as *Ccnd1*, *Apaf1* (*Cammas et al., 2007*), *Myc* (*Jo et al., 2008*), *Fgf2* (*Bonnal et al., 2005*), and *Xiap* (*Lewis et al., 2007*; *Wall and Lewis, 2017*). HnRNPA1 can be methylated by PRMT1 (*Wall and Lewis, 2017*) or PRMT5 (*Gao et al., 2017*; *Holmes et al., 2019*), which regulates the ITAF activity of HnRNPA1. To test whether PRMT5 interacts with HnRNPA1 in granulosa cells, coimmunoprecipitation experiments were conducted. We found that in control granulosa cells HnRNPA1 and PRMT5 were pulled down by antibody against the PRMT5 main binding partner MEP50; conversely, PRMT5 and MEP50 could be pulled down by the HnRNPA1 antibody (*Figure 5E*). Although HnRNPA1 protein expression was not changed between control and *Prmt5^flox/flox*;*Sf1^+/cre* granulosa cells, the level of symmetric dimethylation of HnRNPA1 in *Prmt5^flox/flox*;*Sf1^+/cre* granulosa cells was significantly reduced compared with that in control granulosa cells (*Figure 5E*).

To test whether HnRNPA1 functions during PRMT5-mediated *Wt1* translation, HnRNPA1 was knocked down in granulosa cells via siRNA transfection. Western blot analysis results showed that HnRNPA1 protein levels were significantly decreased with siRNA transfection (*Figure 6A*, *Figure 6—figure supplement 1*). We found that WT1 protein level was increased significantly in granulosa cells after knockdown of HnRNPA1. The decreased WT1 expression in EPZ015666-treated granulosa cells was partially reversed by knockdown of HnRNPA1 (*Figure 6A*, *Figure 6—figure supplement 1*). The luciferase activity of pRWT1F was increased in granulosa cells with HnRNPA1 siRNA treatment and decreased in those with EPZ015666 treatment. The decreased luciferase activity in EPZ015666-treated

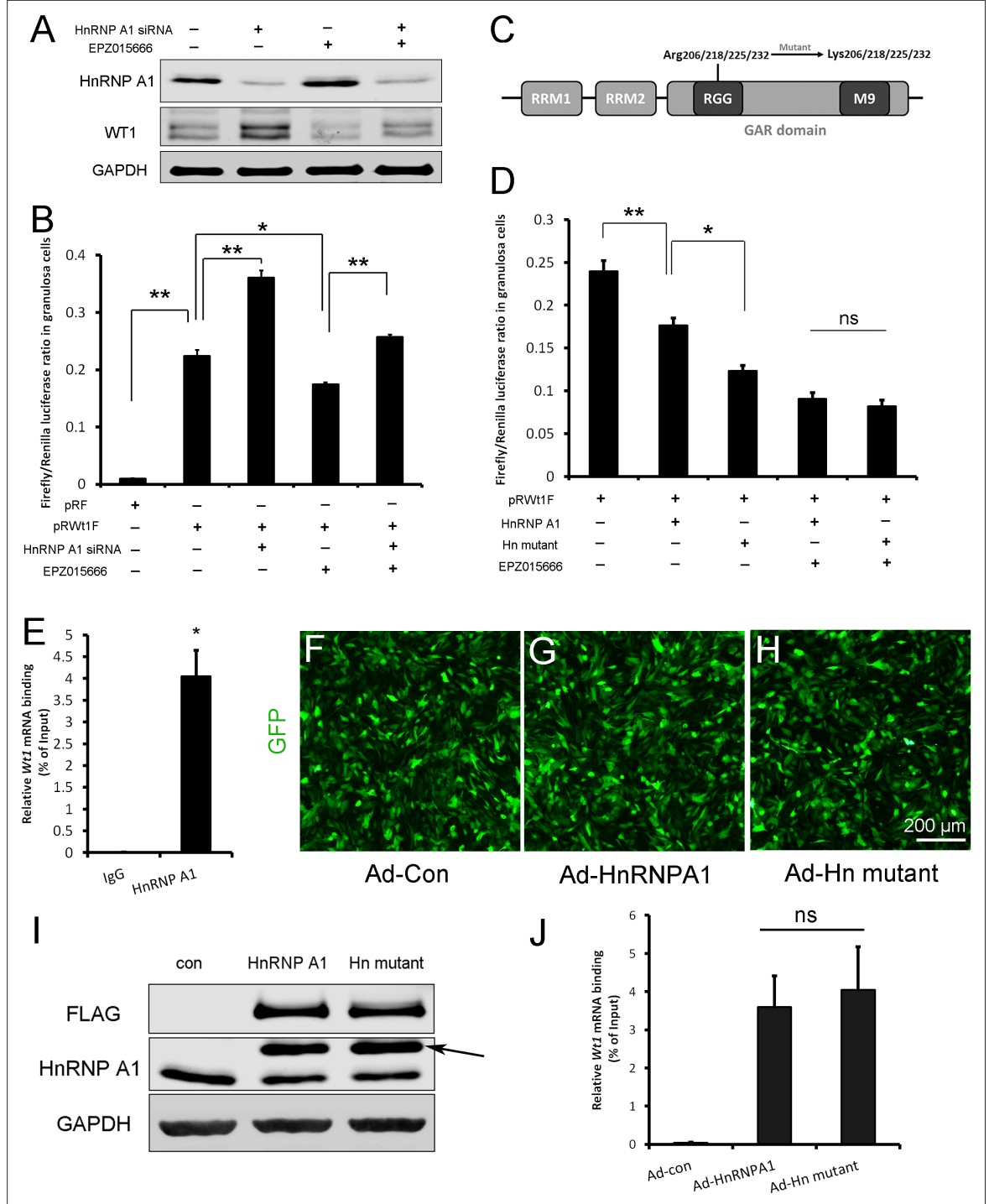

**Figure 6.** *Wt1* internal ribosome entry site (IRES) activity is regulated by PRMT5 via methylation of HnRNPA1. (**A**) Western blot analysis of HnRNPA1 and WT1 in granulosa cells after HnRNPA1 siRNA transfection or EPZ015666 treatment. (**B**) Luciferase activity analysis of pRWT1F in granulosa cells after HnRNPA1 siRNA transfection or EPZ015666 treatment. Isolated granulosa cells were treated with DMSO or EPZ015666 for 4 days. The day granulosa cells isolated is denoted as day 1. On day 2, cells were transfected with control siRNA or siRNA to HnRNPA1. 48 hr later, pRF or pRWT1F were transfected. The luciferase activity of pRWT1F was calculated as the ratio of Firefly luciferase activity to Renilla luciferase activity. (**C**) Schematic diagram of HnRNPA1 protein domains. HnRNPA1 contains two RNA recognition motifs (RRMs). The glycine/arginine-rich (GAR) domain contains an RGG (Arg-Gly-Gly) box and a nuclear targeting sequence (M9). Four arginine residues within the RGG motif were mutated to lysine. (**D**) Luciferase activity analysis of pRWT1F in granulosa cells after EPZ015666 treatment or overexpressing HnRNPA1 or arginine-mutated HnRNPA1. Isolated granulosa cells were treated with DMSO or EPZ015666 for 4 days. On day 3, flag-tagged HnRNPA1 or mutant plasmids were cotransfected with pRWT1F into granulosa cells. 48 hr later, cells were harvested for luciferase activity analysis. (**E**) RNA immunoprecipitation was conducted in granulosa cells using an HnRNPA1

*Figure 6 continued on next page*

*Figure 6 continued*

antibody, and the *Wt1* mRNA pulled down by HnRNPA1 was analyzed with real-time PCR. (**F–H**) Primary granulosa cells were cultured and infected with control, flag-tagged HnRNPA1, or mutant HnRNPA1 (Ad-Hn mutant) adenoviruses. The expression of control and mutant HnRNPA1 was examined by western blot analysis (**I**). (**J**) RNA immunoprecipitation was conducted using a FLAG antibody, and *Wt1* mRNA pulled down by control or mutant HnRNPA1 protein was analyzed with real-time PCR. For (**B, D**) (n = 4) and (**E, J**) (n = 3), the data are presented as the mean ± SEM. *p<0.05. **p<0.01.

The online version of this article includes the following figure supplement(s) for figure 6:

**Source data 1.** Source data for *Figure 6B, D, E and J*.

**Figure supplement 1.** Quantitative analysis of HnRNPA1 and WT1 protein expression in granulosa cells after HnRNPA1 siRNA transfection or EPZ015666 treatment.

**Figure supplement 1—source data 1.** Source data for *Figure 6—figure supplement 1*.

granulosa cells was partially reversed by knockdown of HnRNPA1 (*Figure 6B*). To further confirm the effect of HnRNPA1 on *Wt1* IRES activity, HnRNPA1 was overexpressed in granulosa cells, and we found that *Wt1* IRES activity was significantly decreased (*Figure 6D*). These results indicated that as an ITAF the effect of HnRNPA1 on *Wt1* IRES activity was repressive.

There are five arginine residues in the HnRNPA1 glycine/arginine-rich (GAR) motif, which can be symmetrically or asymmetrically dimethylated by PRMT5 (*Gao et al., 2017*) or PRMT1 (*Rajpurohit et al., 1994*; *Wall and Lewis, 2017*), respectively. R206, R218, R225, and R232 are required for HnRNPA1 ITAF activity (*Gao et al., 2017*; *Wall and Lewis, 2017*). To determine the role of HnRNPA1 arginine methylation in *Wt1* IRES activity, the four arginine residues were mutated to lysines (*Figure 6C*), and flag-tagged HnRNPA1 or mutant plasmids were cotransfected with pRWT1F into granulosa cells. We found that *Wt1* IRES activity was further decreased in granulosa cells overexpressing mutant HnRNPA1 compared to those overexpressing wild-type HnRNPA1 (*Figure 6D*). However, the difference in *Wt1* IRES activity between cells overexpressing mutant HnRNPA1 and cells overexpressing wild-type HnRNPA1 disappeared when the granulosa cells were treated with EPZ015666 (*Figure 6D*). These results indicate that the repressive function of HnRNPA1 on *Wt1* IRES activity is inhibited by PRMT5-mediated arginine symmetric dimethylation.

To test the interaction between HnRNPA1 and *Wt1* mRNA, RNA immunoprecipitation was performed with HnRNPA1 antibody in primary granulosa cells. As shown in *Figure 6E*, *Wt1* mRNA was pulled down by the HnRNPA1 antibody in granulosa cells. Next, granulosa cells were infected with flag-tagged wild-type or arginine-mutant HnRNPA1 adenovirus (*Figure 6F–I*) and RNA immunoprecipitation was conducted with a FLAG antibody. The results showed that mutation of arginines did not affect the interaction between HnRNPA1 and *Wt1* mRNA (*Figure 6J*).

## The upregulation of steroidogenic genes in *Prmt5*<sup>flox/flox</sup>;*Sf1*<sup>+/cre</sup> granulosa cells was repressed by *Wt1* overexpression

To test whether the upregulation of steroidogenic genes in *Prmt5*-deficient granulosa cells is due to downregulation of WT1, granulosa cells from *Prmt5*<sup>flox/flox</sup>;*Sf1*<sup>+/cre</sup> mice were infected with control or GFP-tagged WT1-expressing adenovirus (*Figure 7A and B*). *Wt1* protein (*Figure 7D* arrow, E) and mRNA (*Figure 7C*) levels were dramatically increased in *Prmt5*-deficient granulosa cells after *Wt1* overexpression. We found that the expression of steroidogenic genes was significantly decreased in these cells. These results suggest that the aberrant differentiation of *Prmt5*-deficient granulosa cells can be rescued by WT1.

## Discussion

Protein arginine methylation is one of the most important epigenetic modifications and is involved in many cellular processes. In this study, we found that protein arginine methylation plays important roles in granulosa cell development. The development of ovarian follicles is a dynamic process. The granulosa cells in antral follicles express gonadotropin receptors. Before ovulation, granulosa cells begin to express steroidogenic enzymes that are necessary for progesterone and estradiol synthesis (*Irving-Rodgers et al., 2004*; *Smith et al., 2014*). In this study, we found that *Prmt5*-deficient granulosa cells began to express steroidogenic genes in secondary follicles and the upregulation of the steroidogenic genes in *Prmt5*-deficient granulosa cells was reversed by *Wt1* overexpression, indicating that PRMT5

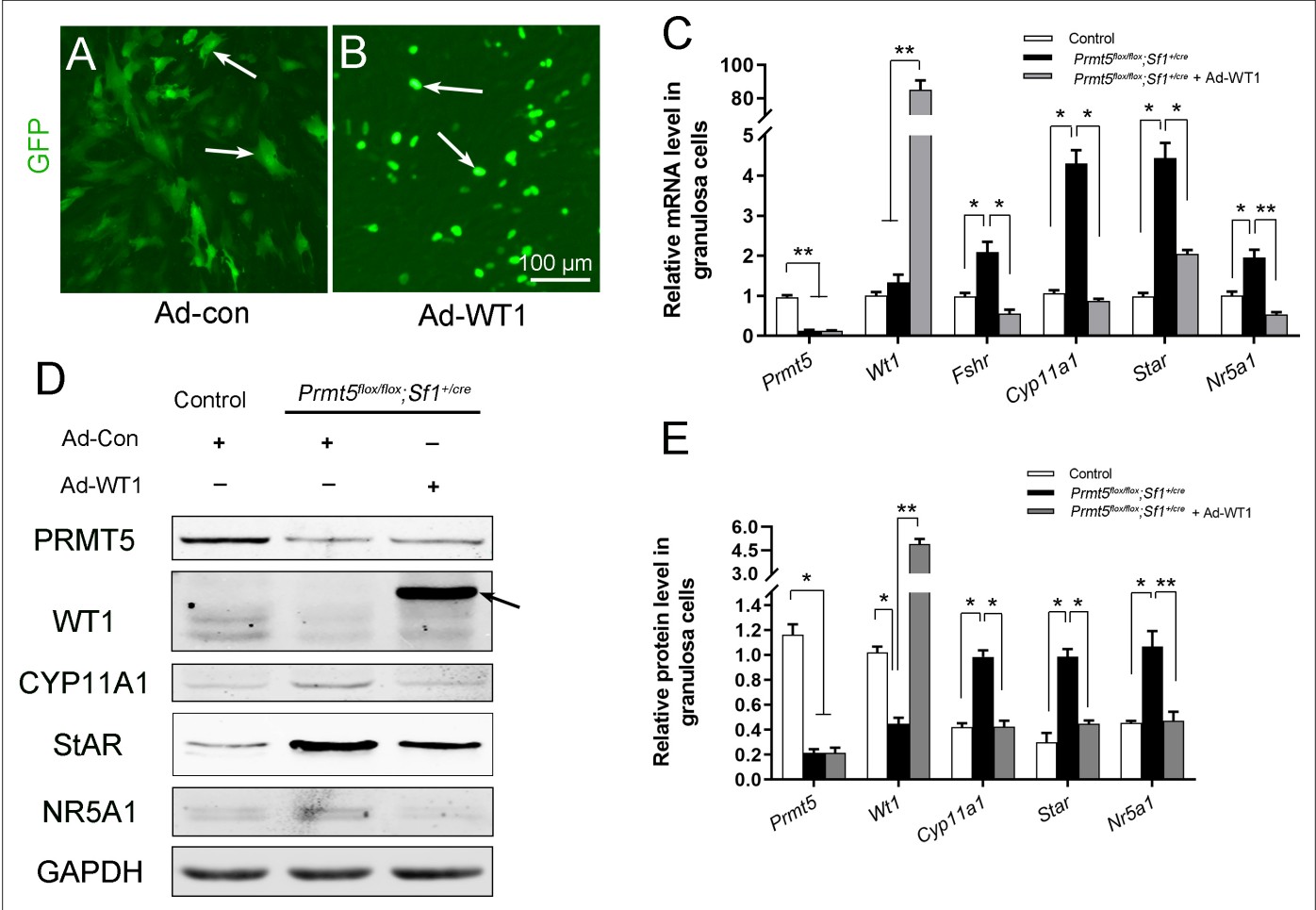

**Figure 7.** The upregulation of steroidogenic genes in *Prmt5^flox/flox^;Sf1^+/cre^* granulosa cells was reversed by *Wt1* overexpression. (**A, B**) Granulosa cells isolated from control and *Prmt5^flox/flox^;Sf1^+/cre^* mice were cultured and infected with control or GFP-fused *Wt1* adenovirus. The expression of steroidogenic genes was examined by RT-qPCR (**C**) and western blot analysis (**D**). The protein expression of three independent experiments in western blot analysis was quantified and normalized to that of GAPDH (**E**). (**C, E**) The data are presented as the mean ± SEM (n = 3). *p<0.05. **p<0.01.

The online version of this article includes the following figure supplement(s) for figure 7:

**Source data 1.** Source data for *Figure 7C and E*.

---

is required for preventing the premature differentiation of granulosa cells via regulation of WT1 expression. Coordinated interaction between granulosa cells and oocytes is required for successful follicle development and production of fertilizable oocytes. The premature luteinized granulosa cells will lose their structural and nutritional support for oocytes, which will lead to follicle growth arrest or atresia at early stages of folliculogenesis.

Nuclear receptor *Sf1* plays a critical role in the regulation of steroid hormone biosynthesis by inducing the expression of steroidogenic enzymes in steroidogenic cells (*Ikeda et al., 1993*). Our previous study demonstrated that WT1 represses *Sf1* expression by directly binding to the *Sf1* promoter region and that inactivation of *Wt1* causes upregulation of *Sf1,* which in turn activates the steroidogenic program (*Chen et al., 2017*). In the present study, the mRNA and protein levels of *Sf1* were significantly upregulated after WT1 loss. Therefore, we speculate that the upregulation of steroidogenic genes in *Prmt5*-deficient granulosa cells is most likely due to the increased expression of *Sf1* gene.

As an important nuclear transcription factor, the function of WT1 in granulosa cell development has been investigated. However, the molecular mechanism that regulates the expression of *Wt1* gene is unknown. In this study, we found that the expression of WT1 at the protein level was dramatically reduced in *Prmt5*-deficient granulosa cells, whereas the mRNA level was not changed, indicating that

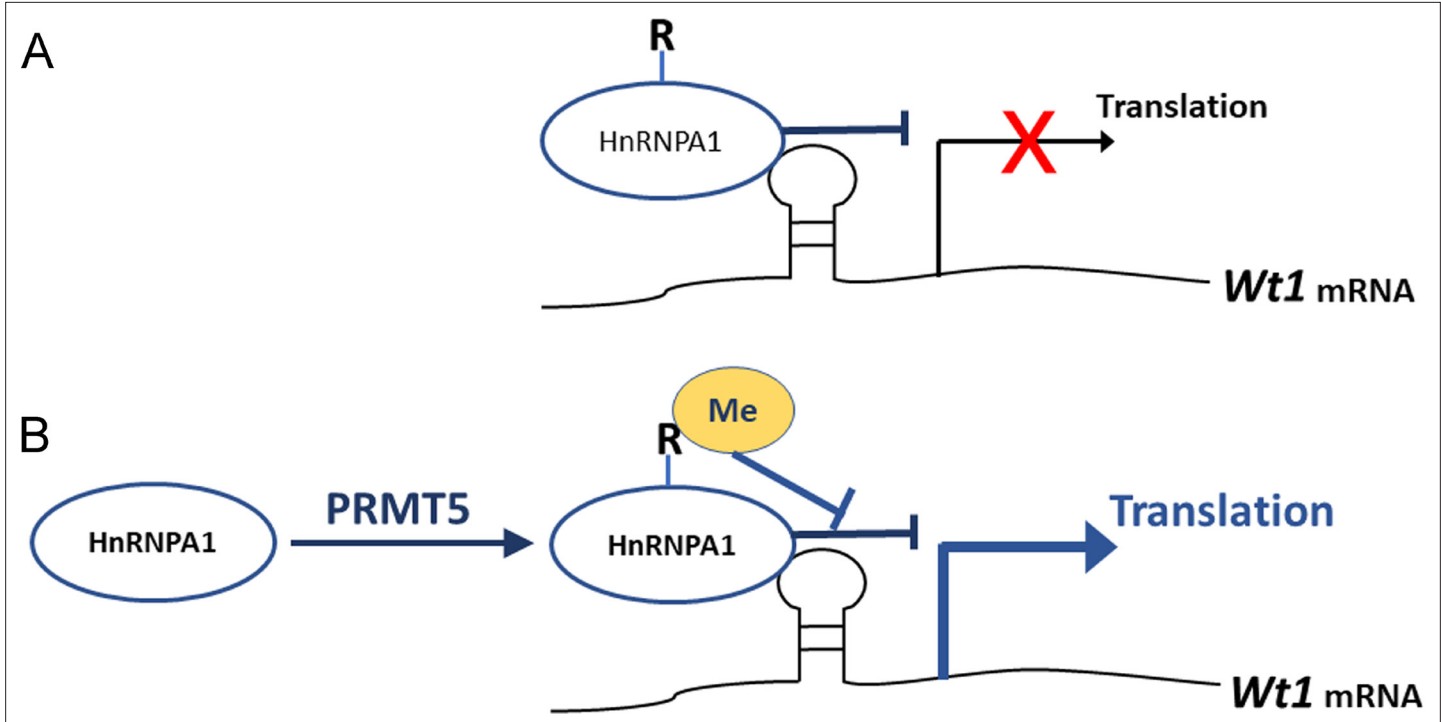

**Figure 8.** Schematic illustration of how PRMT5 regulates *Wt1* mRNA translation. (**A**) As an ITAF, HnRNPA1 binds to *Wt1* mRNA and inhibits the internal ribosome entry site (IRES)-dependent translation of *Wt1*. (**B**) PRMT5 catalyzes symmetric methylation of HnRNPA1, which suppresses the ITAF activity of HnRNPA1 and promotes the translation of *Wt1* mRNA. R: arginine. Me: methylation.

PRMT5 regulates *Wt1* expression at the post-transcriptional level. In our mouse model, *Prmt5* was inactivated in granulosa cells at the early embryonic stage. However, defects in follicle development were not observed until 2 weeks after birth. This outcome probably occurred because *Prmt5* is not expressed in granulosa cells before the development of primary follicles (*Figure 1—figure supplement 1*). During the early stage, *Wt1* expression is also maintained in pregranulosa cells. Therefore, we speculate that there must be other factor(s) involved in regulating *Wt1* expression before primary follicle stage.

More than 100 mRNAs in mammals contain IRES elements in their 5'UTRs (*Jaud et al., 2019*), which are involved in various physiological processes, such as differentiation, cell cycle progression, apoptosis, and stress responses (*Godet et al., 2019*). The 5'UTR sequence of *Wt1* mRNA is highly conserved, with more than 85% homology among the sequences of 29 mammalian species. Our study indicates that the *Wt1* 5'UTR has IRES activity. HnRNPA1 belongs to the HnRNP family, which comprises at least 20 members associated with RNA processing, splicing, transport, and metabolism (*Godet et al., 2019*; *Roy et al., 2017*). As a main ITAF, HnRNPA1 either activates the translation of *Fgf2* (*Bonnal et al., 2005*), *Srebp-1a* (*Damiano et al., 2013*), and *Ccnd1* (*Shi et al., 2005*) or inhibits the translation of *Xiap* (*Lewis et al., 2007*), *Apaf* (*Cammas et al., 2007*), and *Bcl-xl* (*Bevilacqua et al., 2010*). The underlying mechanism by which HnRNPA1 activates some IRESs but suppresses other IRESs is still unknown. HnRNPA1 may compete with other ITAFs for binding or may modify IRES structure and thus regulate IRES activity (*Cammas et al., 2007*; *Lewis et al., 2007*).

It has been reported that the expression of several genes is regulated by PRMT5 at the protein level (*Gao et al., 2017*; *Nicholas et al., 2013*). Gao et al. reported that PRMT5 regulates IRES-dependent translation via methylation of HnRNPA1 in the 293T and MCF-7 cell lines. They found that HnRNPA1 activates the IRES-dependent translation and that methylation of HnRNPA1 facilitates the interaction of HnRNPA1 with IRES mRNA to promote translation (*Gao et al., 2017*). In the present study, we found that *Wt1* IRES activity was repressed by HnRNPA1 (*Figure 8A*) and that the repressive effect of HnRNPA1 was reversed by PRMT5-mediated arginine methylation; thus, *Wt1* IRES-dependent translation was promoted by PRMT5 (*Figure 8B*).

The ITAF activity of HnRNPA1 can be regulated by post-translational modifications (*Godet et al., 2019*). Phosphorylation of HnRNPA1 on serine 199 by Akt inhibits IRES-dependent translation of *c-myc* and *cyclin D1* (*Jo et al., 2008*; *Shi et al., 2005*). Symmetric dimethylation of HnRNPA1 by PRMT5 enhances HnRNPA1 ITAF activity and promotes the translation of target mRNAs (*Gao et al., 2017*). Asymmetric dimethylation of HnRNPA1 by PRMT1 inhibits its ITAF activity (*Wall and Lewis, 2017*). These results suggest that arginine methylation has different effects on the ITAF activity of HnRNPA1 according to different IRESs and cell contexts. Our study demonstrated that HnRNPA1 ITAF activity toward *Wt1* mRNA was repressed by PRMT5-mediated arginine methylation. However, the affinity between HnRNPA1 and *Wt1* mRNA was not affected after mutation of arginine residues, consistent with the findings of a previous study (*Wall and Lewis, 2017*). Therefore, the inhibition of HnRNPA1 ITAF activity by PRMT5 does not occur through changes in the binding of HnRNPA1 to *Wt1* mRNA. Arginine–glycine–glycine (RGG)-motif region is also reported to be involved in mediating the interactions between homo- and heterotypic proteins. It is possible that arginine methylation of HnRNPA1 changes the interactions of HnRNPA1 and its protein partners, which affects the ITAF activity of HnRNPA1 (*Wall and Lewis, 2017*). The underlying mechanism needs further investigation.

Epigenetic modification is involved in numerous cellular processes. However, the functions of epigenetic modification in granulosa cell development have not been well studied. In this study, we demonstrated that *Prmt5* is required for maintenance of granulosa cell identity in follicle development and that inactivation of *Prmt5* causes premature luteinization of granulosa cells. Our study also demonstrates that PRMT5 regulates WT1 expression at the translational level by methylating HnRNPA1. This study provides very important information for better understanding the regulation of gonad somatic cell differentiation.

## Materials and methods

### Key resources table

| Reagent type (species) or resource | Designation | Source or reference | Identifiers | Additional information |
|---|---|---|---|---|
| Genetic reagent (*Mus musculus*) | *Sf1+/cre* | Gift from Prof. Humphrey Hung-Chang Yao | Parker lab | *Bingham et al., 2006* |
| Recombinant DNA reagent | pRF (plasmid) | Gift from Prof. Anne E Willis | Willis lab | *Coldwell et al., 2000* |
| Antibody | Anti-PRMT5 (rabbit polyclonal) | Millipore | Cat# 07-405 | IF (1:200), WB (1:1000) |
| Antibody | Anti-MEP50 (rabbit monoclonal) | Abcam | Cat# ab154190 | WB (1:1000), IP (1 µg/mg protein) |
| Antibody | Anti-WT1 (rabbit monoclonal) | Abcam | Cat# ab89901 | IHC (1:400), IF (1:200), WB (1:1000) |
| Antibody | Anti-FOXL2 (goat polyclonal) | Abcam | Cat# Abcam, ab5096 | IF (1:100), WB (1:800) |
| Antibody | Anti-HnRNPA1 (mouse monoclonal) | Abcam | Cat# ab5832 | WB (1:1000) IP (1 µg/mg protein) |
| Antibody | Anti-SYM10 (rabbit polyclonal) | Millipore | Cat# 07-412 | WB (1:800) |
| Antibody | Anti-CYP11A1 (rabbit polyclonal) | Proteintech | Cat# 13363-1-AP | IF (1:100), WB (1:800) |
| Antibody | Anti-SF1 (rabbit polyclonal) | Proteintech | Cat# 18658-1-AP | IF (1:200), WB (1:800) |
| Antibody | Anti-StAR (rabbit polyclonal) | Santa Cruz | Cat# sc-25806 | IF (1:100), WB (1:800) |
| Antibody | Anti-FLAG (mouse monoclonal) | Sigma-Aldrich | Cat# F1804 | WB (1:1000) RIP (1 µg/mg protein) |
| Antibody | Anti-IgG | Santa Cruz | Cat# sc-2025 | RIP (1 µg/mg protein) |
| Antibody | Anti-3β-HSD (goat polyclonal) | Santa Cruz | Cat# sc-30820 | IF (1:200), WB (1:1000) |

*Continued on next page*

*Continued*

| Reagent type (species) or resource | Designation | Source or reference | Identifiers | Additional information |
|---|---|---|---|---|
| Antibody | FITC-conjugated donkey anti-goat IgG | Jackson | Cat# 705-095-147 | 1:150 |
| Antibody | Cy3-conjugated donkey anti-rabbit IgG | Jackson | Cat# 711-165-152 | 1:300 |
| Commercial assay or kit | Collagenase IV | VETEC | Cat# V900893 | 1 mg/ml |
| Commercial assay or kit | Hyaluronidase | Sigma Aldrich | Cat# SIAL-H3506 | 1 mg/ml |
| Commercial assay or kit | DNase I | AppliChem | Cat# A37780500 | 1 mg/ml |
| Commercial assay or kit | EPZ015666 | MedChemExpress | Cat# HY-12727 | 5 µM |
| Commercial assay or kit | RNeasy Kit | Aidlab | Cat# RN28 | |
| Commercial assay or kit | Dual luciferase reporter assay system | Promega | Cat# E1910 | |
| Commercial assay or kit | siRNA to HnRNPA1 | ThermoFisher Scientific | Cat# S67643, S67644 | |
| Commercial assay or kit | Protein A agarose beads | GE | Cat# 17-5280-01 | |
| Commercial assay or kit | Protein G agarose beads | GE | Cat# 17-0618-01 | |

## Mice

All animal experiments were carried out in accordance with the protocols approved by the Institutional Animal Care and Use Committee (IACUC) of the Institute of Zoology, Chinese Academy of Sciences (CAS; SYXK 2018-0021). All mice were maintained on a C57BL/6;129/SvEv mixed background. *Prmt5$^{flox/flox}$;Sf1$^{+/cre}$* female mice were obtained by crossing *Prmt5$^{flox/flox}$* mice with *Prmt5$^{+/flox}$;Sf1$^{+/cre}$* mice. *Prmt5$^{flox/flox}$* and *Prmt5$^{+/flox}$* female mice were used as controls.

For breeding experiment, both control and *Prmt5$^{flox/flox}$;Sf1$^{+/cre}$* female mice were crossed with wild-type male mice when they reached 8 weeks of age. Each pair was kept in a cage for 4 months. The number of pups delivered during this period was counted.

## Plasmid and adenovirus

The dicistronic construct pRF was a generous gift from Professor Anne Willis, University of Cambridge. pRWT1F, pRCCND1F, and pRWT1-RevF were constructed by inserting the mouse *Wt1* 5′UTR, human *Ccnd1* 5′UTR, or mouse *Wt1* 5′UTR in reverse orientation into EcoRI and NcoI sites of the pRF vector. Mouse *Wt1* 5′UTR and human *Ccnd1* 5′UTR sequence were amplified by PCR. The primers amplifying the whole transcript of pRF binding to the 5′ end of Renilla and 3′ end of Firefly ORF: pRF-F: GCCACCATGACTTCGAAAGTTTATGA; pRF-R: TTACACGGCGATCTTTCCGC. FLAG-tagged HnRNPA1 and mutant plasmids were generated by inserting the coding sequence and a mutant sequence of mouse HnRNPA1, respectively, into NheI and BamHI sites of the pDC316-mCMV-ZsGreen-C-FLAG vector. Adenoviruses containing WT1 coding sequence, HnRNPA1, or the mutant sequence were generated using the Gateway Expression System (Invitrogen). The primers used for constructing the plasmids are as follows:

| Plasmid symbol | Primer 5′ to 3′ |
|---|---|
| pRWT1F-forward pRWT1F-reverse | CCGGAATTCTGTGTGAATGGAGCGGCCGAGCAT CTAGCCATGGGATCGCGGCGAGGAGGCG |
| pRWT1-RevF-forward | CTAGCCATGGTGTGTGAATGGAGCGGCCGAGCAT |
| pRWT1-RevF-reverse | CCGGAATTCGATCGCGGCGAGGAGGCG |
| pRCCND1F-forward | GCTGAATTCCACACGGACTACAGGGGAGTTTT |
| pRCCND1F-reverse pRWT1F −268 to −158-forward | CGGCCATGGGGGCTGGGGCTCTTCCTGGGC CCGGAATTCTGTGTGAATGGAGCGGCCGAG |

*Continued on next page*

*Continued*

| Plasmid symbol | Primer 5' to 3' |
|---|---|
| pRWT1F –268 to –158-reverse | CTAGCCATGGACTCCTTACCCCAGCTGCCT |
| pRWT1F –198 to –58-forward | CCGGAATTCTTTGGGAAGCTGGGGGCAG |
| pRWT1F –198 to –58-reverse<br>pRWT1F –105 to 1-forward pRWT1F<br>–105 to 1-reverse<br>HnRNPA1-forward<br>HnRNPA1-reverse | CTAGCCATGGGGTGGGTGAATGAGTGGGT<br>CCGGAATTCTCCCCCTTTCCTTTTCCCGCCC<br>CTAGCCATGGGATCGCGGCGAGGAGGCG<br>CTAGCTAGCCACCATGTCTAAGTCCGAGTCTCCCAAGGA<br>CGCGGATCCGAACCTCCTGCCACTGCCATAGCTA |

## Isolation of granulosa cells, transient transfection, infection, and luciferase assay

Granulosa cells were isolated from mice at 16–18 days old. After mechanical dissection, ovaries were cut into several parts and incubated in PBS containing 1 mg/ml collagenase IV (VETEC, V900893) in a water bath with circular agitation (85 rpm) for 5 min at 37 °C. Follicles were allowed to settle and washed in PBS. The supernatant were discarded. A second enzyme digestion was performed in PBS containing 1 mg/ml collagenase IV, 1 mg/ml hyaluronidase (SIGMA, SIAL-H3506), 0.25% Trypsin, and 1 mg/ml DNase I (AppliChem, A37780500) for 15 min. FBS was added to stop the digestion and cell suspension was filtered through a 40 μm filter. Cells were centrifuged, washed, and then plated in 24-well plate in DMEM/F12 supplemented with 5% FBS. For EPZ015666 treatment, granulosa cells were incubated in the medium with the addition of 5 μM EPZ015666 (MedChemExpress, HY-12727) for 4–5 days. When cells were approximately 70% confluent, granulosa cells were transfected with plasmids or infected with adenovirus according to the experiments. At the end of culture, cells were lysed for RT-qPCR, western blot analysis, or luciferase activity analysis using a dual luciferase reporter assay system (Promega, E1910).

Control siRNA or siRNA to HnRNPA1 was purchased from ThermoFisher (S67643, S67644) and transfected into granulosa cells with Lipofectamine 3000 transfection reagent without P3000. 48 hr later, pRF or pRWT1F were transfected and luciferase activities were measured the following day.

## In vitro ovarian follicle culture

Follicles were dissected and cultured as previously described (*Gao et al., 2014*). Briefly, ovaries of 14-day-old mice were dissected aseptically using the beveled edges of two syringe needles. Follicles with 2–3 layers of granulosa cells, a centrally placed oocyte, an intact basal membrane, and attached theca cells were selected and cultured individually in 20 μl droplets of culture medium (αMEM supplemented with 5% FBS, 1% ITS, and 100 mIU/ml recombinant FSH). The culture were maintained in 37 °C and 5% $CO_2$ in air. The medium was replaced every other day. The histology of the follicles was recorded daily under a microscope.

## Coimmunoprecipitation

Granulosa cells isolated from mice at 16–18 days old were cultured in 10 cm dishes and lysed with lysis buffer (50 mM Tris–HCl [pH 7.5], 150 mM NaCl, 1 mM EDTA, 1% Nonidet P-40) supplemented with protease inhibitors cocktail (Roche) and 1 mM PMSF. 1 mg of protein were first pre-cleared with protein A/G agarose beads (GE, 17-0618-01, 17-5280-01) for 1 hr at 4 °C, then incubated with HnRNPA1 antibody (Abcam, ab5832), or MEP50 antibody (Abcam, ab154190) for 4 hr at 4 °C. Then protein A and G agarose beads were added and incubated overnight. The immunoprecipitates were washed four times in lysis buffer supplemented with cocktail and PMSF, resolved in loading buffer, incubated for 5 min at 95 °C, and then analyzed by western blotting. The antibodies used in western blotting include PRMT5 (Millipore, 07-405), SYM10 (Millipore, 07-412), HnRNPA1 (Abcam, ab5832), and MEP50 (Abcam, ab154190).

## Western blot analysis

Granulosa cells were washed with PBS, lysed with RIPA buffer (50 mM Tris–HCl [pH 7.5], 150 mM NaCl, 1% NP-40, 0.1% SDS, 1% sodium deoxycholate, 5 mM EDTA) supplemented with protease

inhibitors cocktail (Roche) and 1 mM PMSF. Equal amounts of total protein were separated by SDS/PAGE gels, transferred to nitrocellulose membrane, and probed with the primary antibodies. The images were captured with the ODYSSEY Sa Infrared Imaging System (LI-COR Biosciences, Lincoln, NE). Densitometry was performed using ImageJ software. The protein expression was normalized to that of GAPDH. Blots are representative of three independent experiments. The antibodies used were PRMT5 (Millipore, 07-405), MEP50 (Abcam, ab154190), WT1 (Abcam, ab89901), FOXL2 (Abcam, ab5096), CYP11A1 (Proteintech, 13363-1-AP), StAR (Santa Cruz, sc-25806), SF1 (Proteintech, 18658-1-AP), and FLAG (Sigma, F1804).

## RNA immunoprecipitation

Granulosa cells were isolated from mice at 16–18 days old and cultured in 10 cm dishes. The cells were then lysed with RIP buffer (50 mM Tris-HCl [pH 7.5], 150 mM NaCl, 5 mM EDTA, 1% NP-40, 0.5% sodium deoxycholate) supplemented with protease inhibitor cocktail and 200 U/ml RNase inhibitor. 5% of the cell lysate supernatants were used as the input and the remaining were incubated with 1.5 µg of IgG (mouse, Santa Cruz, sc-2025), HnRNPA1 antibody (Abcam, ab5832), or FLAG antibody (Sigma, F1804) for 4 hr at 4 °C. Then protein A and G agarose beads were added to immunoprecipitate the RNA/protein complex. The conjugated beads were thoroughly washed with lysis buffer (50 mM Tris–HCl [pH 7.5], 500 mM NaCl, 5 mM EDTA, 1% NP-40, 0.5% sodium deoxycholate) supplemented with cocktail and 200 U/ml RNase inhibitor. Bound RNA was extracted using a RNeasy Kit and analyzed with RT-qPCR analysis.

## Real-time RT-PCR

Total RNA was extracted using a RNeasy Kit (Aidlab, RN28) in accordance with the manufacturer's instructions. 1 µg of total RNA was used to synthesize first-strand cDNA (Abm, G592). cDNAs were diluted and used for the template for real-time SYBR Green assay. *Gapdh* was used as an endogenous control. All gene expression was quantified relative to *Gapdh* expression. The relative concentration of the candidate gene expression was calculated using the formula $2^{-\Delta\Delta CT}$. Real-time RT-PCR primers are as follows:

| Gene symbol | RT forward primer 5′ to 3′ | RT reverse primer 5′ to 3′ |
| --- | --- | --- |
| *Prmt5* | TGGTGGCATAACTTTCGGACT | TCCAAGCCAGCGGTCAAT |
| *Wt1* | CAAGGACTGCGAGAGAAGGTTT | TGGTGTGGGTCTTCAGATGGT |
| *Hsd3b1* | CTCAGTTCTTAGGCTTCAGCAATTAC | CCAAAGGCAAGATATGATTTAGGA |
| *Cyp11a1* | CCAGTGTCCCCATGCTCAAC | TGCATGGTCCTTCCAGGTCT |
| *Cyp17a1* | GCCCAAGTCAAAGACACCTAAT | GTACCCAGGCGAAGAGAATAGA |
| *StAR* | CCGGAGCAGAGTGGTGTCA | CAGTGGATGAAGCACCATGC |
| *Sf1* | CCCAAGAGTTAGTGCTCCAGT | CTGGGCGTCCTTTACGAGG |
| *Foxl2* | ACAACACCGGAGAAACCAGAC | CGTAGAACGGGAACTTGGCTA |
| *Fshr* | ATGTGTTCTCCAACCTACCCA | GCTGGCAAGTGTTTAATGCCTG |
| *Gapdh* | GTCATTGAGAGCAATGCCAG | GTGTTGCTACCCCCAATGTG |
| *Renilla* | CGTGGAAACCATGTTGCCATCAA | ACGGGATTTCACGAGGCCATGATA |
| *Firefly* | GGTTCCATCTGCCAGGTATCAGG | CGTCTTCGTCCCAGTAAGCTATG |
| *Cyp19a1* | AACCCCATGCAGTATAATGTCAC | AGGACCTGGTATTGAAGACGAG |

## Immunohistochemistry and immunofluorescence analysis

Immunohistochemistry procedures were performed as described previously (*Gao et al., 2006*). Stained sections were examined with a Nikon microscope, and images were captured by a Nikon DS-Ri1 CCD camera. For immunofluorescence analysis, the 5 µm sections were incubated with 5% BSA in 0.3% Triton X-100 for 1 hr after rehydration and antigen retrieval. The sections were then incubated with the primary antibodies for 1.5 hr and the corresponding FITC-conjugated donkey anti-goat IgG (1:150, Jackson ImmunoResearch, 705-095-147) and Cy3-conjugated donkey anti-rabbit IgG (1:300, Jackson ImmunoResearch, 711-165-152) for 1 hr at room temperature. The following primary antibodies were used: WT1 (Abcam, ab89901), FOXL2 (Abcam, ab5096), CYP11A1 (Proteintech, 13363-1-AP), SF1

(Proteintech, 18658-1-AP), and 3β-HSD (Santa Cruz, sc-30820). After being washed three times in PBS, the nuclei were stained with DAPI. The sections were examined with a confocal laser scanning microscope (Carl Zeiss Inc, Thornwood, NY).

For follicle counting analysis, whole ovaries from control and *Prmt5$^{flox/flox}$;Sf1$^{+/cre}$* female mice at 2, 3, 4, and 5 weeks of age were serially sectioned at 5 μm thickness (n = 3/time point/genotype), and follicles were counted on every five sections.

## Statistical analysis

All experiments were repeated at least three times. 3–5 mice for each genotype at each time point were used for immunostaining or quantitative experiments. For immunostaining, one representative picture of similar results from 3 to 5 mice for each genotype at each time point is presented. The quantitative results are presented as the mean ± SEM. All granulosa cell culture experiments were repeated at least three times by using three different cell preparations. Statistical analyses were conducted using GraphPad Prism version 9.0.0. Unpaired two-tailed Student's t-tests were used for comparison between two groups. For three or more groups, data were analyzed using one-way ANOVA. p-Values<0.05 were considered to indicate significance.

## Acknowledgements

We thank Prof. Anne E Willis (University of Cambridge) for her generous gift of the dicistronic contruct pRF. We thank Prof. Humphrey Hung-Chang Yao (NIEHS/NIH) for the *Sf1$^{+/cre}$* mice. This work was supported by the National Key R&D Program of China (2018YFC1004200, 2018YFA0107700); Strategic Priority Research Program of the Chinese Academy of Sciences (XDB19000000); the National Science Fund for Distinguished Young Scholars (81525011); and the National Natural Science Foundation of China (31970785, 31601193, and 31671496).

## Additional information

### Funding

| Funder | Grant reference number | Author |
| --- | --- | --- |
| Ministry of Science and Technology of the People's Republic of China | 2018YFC1004200 | Min Chen |
| Ministry of Science and Technology of the People's Republic of China | 2018YFA0107700 | Fei Gao |
| Chinese Academy of Sciences | XDB19000000 | Fei Gao |
| National Science Fund for Distinguished Young Scholars | 81525011 | Fei Gao |
| National Natural Science Foundation of China | 31970785 | Fei Gao |
| National Natural Science Foundation of China | 31601193 | Fei Gao |
| National Natural Science Foundation of China | 31671496 | Fei Gao |

The funders had no role in study design, data collection and interpretation, or the decision to submit the work for publication.

### Author contributions

Min Chen, Conceptualization, Formal analysis, Funding acquisition, Supervision, Writing – original draft; Fangfang Dong, Data curation, Methodology, Writing – review and editing; Min Chen, Changhuo Cen, Visualization, Writing – review and editing; Zhiming Shen, Haowei Wu, Investigation,

Writing – review and editing; Xiuhong Cui, Validation, Writing – review and editing; Shilai Bao, Writing – review and editing; Fei Gao, Conceptualization, Funding acquisition, Project administration, Writing – original draft

### Author ORCIDs
Min Chen  http://orcid.org/0000-0001-5074-2174
Min Chen  http://orcid.org/0000-0001-6577-6705
Fei Gao  http://orcid.org/0000-0002-4029-6411

### Ethics
This study was performed in strict accordance with the recommendations in the Guide for the Care and Use of Laboratory Animals of Institute of Zoology, Chinese Academy of Sciences (CAS). All animal experiments were carried out in accordance with the protocols approved by the Institutional Animal Care and Use Committee at the Institute of Zoology, Chinese Academy of Sciences (CAS) (Permit Number: SYXK 2018-0021).

### Decision letter and Author response
Decision letter https://doi.org/10.7554/eLife.68930.sa1
Author response https://doi.org/10.7554/eLife.68930.sa2

---

## Additional files

### Supplementary files
• Transparent reporting form

### Data availability
Our work did not generate any datasets or use any previously published datasets. Source data for Figures 4, 5, 6, 7 and figure supplement of Figures 1, 5, 6 have been provided.

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
