## [Decision Letter]

**Acceptance summary:**

Chen and colleagues analyze the function of methyltransferase PRMT5 in ovarian granulosa cells of growing follicles using a conditional mutation that ablates this protein in the granulosa cells. The paper is well-written and of high quality, and targets a major question in developmental biology and reproductive biology.

**Decision letter after peer review:**

Thank you for sending your article entitled "PRMT5 regulates ovarian follicle development by facilitating Wt1 translation" for peer review at *eLife*. Your article is being evaluated by 3 peer reviewers, one of whom is a member of our Board of Reviewing Editors, and the evaluation is being overseen by Marianne Bronner as the Senior Editor. The following individual involved in review of your submission has agreed to reveal their identity: Richard Freiman (Reviewer #3)

Summary:

All the Reviewers have agreed that your manuscript is very strong and provides new information with regard to PRMT5 role in mouse ovarian development. The Reviewers also raised some concerns with regard to the classification/nomenclature of the follicles, lack of fertility data, and lack of aromatase gene expression and critical methylation data and explanation with regard to specificity of the inhibitor used. More detailed comments of each Reviewer are provided below.

*Reviewer #1:*

Epigenetic regulation of ovarian granulosa cells is not well studied. The Authors have knocked out an important regulatory gene Prmt5 specifically in granulose cells and studied the consequences on ovarian development.

While there are several strengths in the paper, the Discussion section is poorly written and some key data with regard to important enzymes in granulosa cell function and female fertility data are missing.

In this manuscript, the Authors have generated mice with a granulosa cell-specific knockout of protein arginine methyltransferase 5 (PRDM5). They show that mutant female mice are infertile and demonstrate that ovarian granulosa cells exhibit premature luteinization with elevated SF1 and steroidogenic enzymes as a result of suppressed WT1. Overexpression of WT1 in mutant cells lacking Prmt5 rescues steroidogenic enzyme encoding gene expression. Further studies identified that Wt1 mRNA is regulated at the translational level by methylation of HnRNPA1. These are well carried out studies and provide new evidence for granulosa cell differentiation at an epigenetic level. PRMT5 regulation of IRES-dependent translation via methylation of hnRNP A1 has already been demonstrated by the Authors previously. There are several other concerns described below that could be addressed to improve this otherwise solid manuscript.

– Several places in the text and figure legends, the histology of ovaries is described as morphology. This needs to be fixed.

– Figure 1: what is the arrow in panel K referring to? The granulosa cell defects in individual follicles are very difficult to appreciate. Please enlarge the region of interest or include a high magnification image.

– Where relevant, the Authors have mentioned "granulosa cell-to-steroidogenic cell transformation" granulosa cells are also steroidogenic cells. Change the text uniformly to "premature luteinized granulosa cells or granulosa cells prematurely luteinized", where relevant.

– There is no clear explanation provided for WT1 repression of steroidogenic enzyme genes in normal progression of folliculogenesis. Is it SF1 mediated as well?

– The most important enzyme for granulosa cell function is Cyp19a1 or aromatase. There is no data provided for this gene expression.

– There are no data provided with regard to fertility of Prmt5 cKO female mice.

– The Discussion as described in lines 436 – 479 reads mostly like Introduction or Results sections. The Discussion section needs to be modified/condensed.

*Reviewer #2:*

In this manuscript from Chen and colleagues, the authors analyze the function of methyltransferase PRMT5 in ovarian granulosa cells of growing follicles using a conditional mutation that ablates this protein expression in the granulosa cells. In Prmt5-knockout mice, follicle development was arrested with disorganized granulosa cells, leading to female infertility. They found that the premature differentiated granulosa cells were detached from oocytes caused by reduced expression of WT1 and the increased expression of steroidogenesis-related genes. Mechanism studies revealed that PRMT5 facilitated IRES-dependent translation of Wt1 mRNA by methylating HnRNPA1. They also performed rescue experiments and showed that to overexpress WT1 in Prmt5-deficient granulosa cells could rescue the aberrant differentiation.

WT1 is a well-known protein in follicle development and this work is a major advance in understanding the role of post-translational arginine methylation in regulating WT1 expression in granulosa cell differentiation and follicle development. They used an array of overall high-quality experiments using mouse granulosa cells including co-immunoprecipitation, RNA immunoprecipitation to illustrate that Wt1 IRES-dependent translation was promoted by PRMT5. Below there is a list of comments that I think the authors should address before considering publication.

1. Line 132: I find the description of the follicle composition in the different ages and mutant mice are very confusing, lacking crucial quantification of the follicles. I think authors should provide:

(1). Detailed definition that they used for different kinds of follicles (primordial, primary, secondary, antral, preantral, …) with representative pictures;

(2). Provide a quantification of follicles at different stage presented in control and mutant animals at different times (2 weeks, 3 weeks, …) in Figure 1.

2. Line 149: Please provide more details about the fertility test. I wonder when the follicles were all lost in mutant mice. Figure S4 showed that the apoptosis and proliferation of granulosa cells was not changed in Prmt5flox/flox; Sf1-Cre mice at P14 and P18.

3. Please specify clearly which band is WT1 and *FOXL2* in Figure 5A.

4. Please provide a little bit more background about the Sf1 Cre mice and point out when and where the Cre recombinase expressed.

5. Please provide more information about how quantification of protein levels on WBs is done.

6. For Figure 6D-F, the authors use IgG as negative control, and I think the Prmt5flox/flox; Sf1-Cre granulosa cells are a better control for the co-immunoprecipitation. In Figure 6F, the western blot results of HnRNPA1 and SYM10 are from the same membrane? I wonder what the expression of HnRNPA1 and SYM10 are in Prmt5-deficient granulosa cells.

7. It is crucial to analyze the methylation status of HnRNPA1 in P14 and P18 Prmt5-deficient granulosa cells.

8. Line 585, A "secong" enzyme?

9. Line 537, the authors mentioned that "the inhibition of HnRNPA1 ITAF activity by PRMT5 does not occur through changes in the binding of HnRNPA1 to Wt1 mRNA", please add some discussions about the possible regulation mechanism between the HnRNPA1 and WT1.

*Reviewer #3:*

In their manuscript entitled "PRMT5 regulates ovarian follicle development by facilitating Wt1Translation" Chen and colleagues test the hypothesis that granulosa cell-specific expression of PRMT5 is required for ovarian follicle development in the mouse. To answer this question, the authors successfully used a Sf1-CRE driver to selectively disrupt a floxxed allele of PRMT5 in granulosa cells. The manuscript is roughly divided into two main sections. The first section addresses ovarian follicle development defects associated with the granulosa cell-specific ablation of PRMT5 and the second section defines the molecular details associated with the gene expression changes.

It appears that the authors have completed and presented a series of important experiments that are novel and of potential high impact. The manuscript is well written and clear and the data are of excellent quality. The study is unique in that it includes highly complicated set of integrated in vivo and in vitro studies to identify molecular mechanisms underlying the regulation of ovarian follicle development. The major findings and conclusions are largely supported by the data that is presented in the manuscript and these conclusions are highly impactful to the field of regulation of gene expression underlying mammalian reproduction and development.

One important concern is that although many clear data are presented, the quantitative and reproducible nature of this data remains questionable. In many cases it is unclear in how many biological or experimental replicates the data is presented from and if sufficient replicates were used to make these important conclusions. This is true for the in vivo mouse data where clear examples of the effects are shown without the necessary quantification or sufficient replicate numbers to make the strong conclusions stated. This is especially true for assessing ovarian follicle numbers between individuals that do vary even within genotypes. In similar fashion, the manuscript hypothesizes the existence of a novel IRES in the 5'UTR of WT1 presents strong evidence for it but never attempts to identify this IRES. The absence of some minor controls and obvious tests of their hypothesis is also a bit concerning. For example, while the rescue experiments with the cDNA for WT1 are really exciting, but this construct missing the 5'UTR of WT1 should be insensitive to the PRMT5 inhibitor but this is not tested. The absence of these controls or additional experimental manipulations detracts from the potential high impact of these findings.

While the granulosa cell targeting of PRMT5 looks successful in Figure S2, Figure 1 is highly descriptive in nature and not sufficiently quantitative. Numbers of each follicle type from multiple biological replicates and fertility status of multiple replicates of each genotype must be completed over a significant period of time. Mating data concluding infertility is nowhere to be found in the manuscript. Reduction of WT1 in the PRMT5 KO at P18 is interesting, but how many replicates of each genotype were assessed and can the staining be analyzed in a more quantitative nature in Figure 2. Figure 4 seems redundant given the fact that follicle structures are disorganized in vivo unless a specific cellular process is examined. For example, does steroid biosynthesis increase in the cKO cultures vs. controls? There are a few key additions possibly already completed or that need to be completed that would enhance the strength of the stated developmental conclusions.

Similar details of the molecular analyses are missing as well that would increase the strength of these conclusions. Is the inhibitor specific for PRMT5 or are other arginine methyltransferases inhibited that could affect the stated conclusions in the manuscript. In figure 6C, the CCND1 IRES is also predicted to be affected by this inhibitor but this is not tested or shown. What is the source of the input differences in figure 6D and E? In 6D there is relatively equivalent levels of all three proteins and in E HNRNPA1 is highly reduced and does this affect the outcome of the IP? How many times was the experiment shown in 6F completed and what is the potential variability observed between replicates? In figure 7A, is there a significant difference in WT1 expression levels or is there a relative increase? Presenting and/or quantifying multiple replicates of each manipulation might help distinguish between these two possibilities. In figure 8, is the adenovirus expression of WT1 cDNA without 5'UTR insensitive to the PRMT5 inhibitor?

---

## [Author Response]

Reviewer #1:Epigenetic regulation of ovarian granulosa cells is not well studied. The Authors have knocked out an important regulatory gene Prmt5 specifically in granulose cells and studied the consequences on ovarian development.While there are several strengths in the paper, the Discussion section is poorly written and some key data with regard to important enzymes in granulosa cell function and female fertility data are missing.In this manuscript, the Authors have generated mice with a granulosa cell-specific knockout of protein arginine methyltransferase 5 (PRDM5). They show that mutant female mice are infertile and demonstrate that ovarian granulosa cells exhibit premature luteinization with elevated SF1 and steroidogenic enzymes as a result of suppressed WT1. Overexpression of WT1 in mutant cells lacking Prmt5 rescues steroidogenic enzyme encoding gene expression. Further studies identified that Wt1 mRNA is regulated at the translational level by methylation of HnRNPA1. These are well carried out studies and provide new evidence for granulosa cell differentiation at an epigenetic level. PRMT5 regulation of IRES-dependent translation via methylation of hnRNP A1 has already been demonstrated by the Authors previously. There are several other concerns described below that could be addressed to improve this otherwise solid manuscript.– Several places in the text and figure legends, the histology of ovaries is described as morphology. This needs to be fixed.

As suggested by the reviewer, “the morphology” has been revised as “the histology” in the revised manuscript.

– Figure 1: what is the arrow in panel K referring to? The granulosa cell defects in individual follicles are very difficult to appreciate. Please enlarge the region of interest or include a high magnification image.

The arrow in panel K refers to the antral follicle in control ovaries which was almost absent in *Prmt5*-deficient ovaries. As suggested by the reviewer, high magnification images for Figure 1E-L have been included in the revised manuscript.

– Where relevant, the Authors have mentioned "granulosa cell-to-steroidogenic cell transformation" granulosa cells are also steroidogenic cells. Change the text uniformly to "premature luteinized granulosa cells or granulosa cells prematurely luteinized", where relevant.

Thank you very much for the suggestion. We have modified the description in the revised manuscript.

– There is no clear explanation provided for WT1 repression of steroidogenic enzyme genes in normal progression of folliculogenesis. Is it SF1 mediated as well?

Thank you very much for the comments. During follicle development, WT1 is abundantly expressed in the granulosa cells of primordial, primary and secondary follicles, and dramatically reduced in antral follicles. By contrast, SF1 expression can not be detected in the granulosa cells of primordial, primary and secondary follicles, and it is upregulated in antral follicles. Our previous studies have found that inactivation of *Wt1* during embryonic stage resulted in up-regulation of steroidogenic enzymes in pre-granulosa cells and Sertoli cells. We also demonstrated that WT1 can directly bind to the *Sf1* promoter region to repress *Sf1* expression (Development, 2017, doi: 10.1242/dev.144105). Given the fact that *Sf1* plays a critical role in the development of steroidogenic cells and induction of steroidogenic gene expression, and significantly increased expression of *Sf1* in *Prmt5*-deficient granulosa cells in the present study, we speculate that WT1 represses steroidogenic enzyme genes expression in granulosa cells probably also by inhibiting *Sf1* expression. But this needs further investigation.

– The most important enzyme for granulosa cell function is Cyp19a1 or aromatase. There is no data provided for this gene expression.

Thank you very much for the suggestion. Actually, we have tried different commercial antibodies. Unfortunately, all of them did not work. Therefore, we analyzed mRNA level of *Cyp19a1* by RT-qPCR. *Cyp19a1* mRNA level was significantly increased in *Prmt5*-deficient granulosa cells. *Cyp19a1* is expressed in granulosa cells of ovary, and its expression increases when follicles develop to antral and preovulatory stage. Therefore, the increased mRNA level in *Cyp19a1* is consistent with our observation that *Prmt5*-deficient granulosa cells prematurely luteinized. The results have been added to Figure 4C of revised manuscript.

– There are no data provided with regard to fertility of Prmt5 cKO female mice.

As suggested by the reviewer, the results of fertility test have been added to the revised manuscript as Figure 1—figure supplement 3.

– The Discussion as described in lines 436 – 479 reads mostly like Introduction or Results sections. The Discussion section needs to be modified/condensed.

Thank you very much for the comments, we have modified the Discussion section as suggested by reviewer.

Reviewer #2:In this manuscript from Chen and colleagues, the authors analyze the function of methyltransferase PRMT5 in ovarian granulosa cells of growing follicles using a conditional mutation that ablates this protein expression in the granulosa cells. In Prmt5-knockout mice, follicle development was arrested with disorganized granulosa cells, leading to female infertility. They found that the premature differentiated granulosa cells were detached from oocytes caused by reduced expression of WT1 and the increased expression of steroidogenesis-related genes. Mechanism studies revealed that PRMT5 facilitated IRES-dependent translation of Wt1 mRNA by methylating HnRNPA1. They also performed rescue experiments and showed that to overexpress WT1 in Prmt5-deficient granulosa cells could rescue the aberrant differentiation.WT1 is a well-known protein in follicle development and this work is a major advance in understanding the role of post-translational arginine methylation in regulating WT1 expression in granulosa cell differentiation and follicle development. They used an array of overall high-quality experiments using mouse granulosa cells including co-immunoprecipitation, RNA immunoprecipitation to illustrate that Wt1 IRES-dependent translation was promoted by PRMT5. Below there is a list of comments that I think the authors should address before considering publication.1. Line 132: I find the description of the follicle composition in the different ages and mutant mice are very confusing, lacking crucial quantification of the follicles. I think authors should provide:(1). Detailed definition that they used for different kinds of follicles (primordial, primary, secondary, antral, preantral, …) with representative pictures;

We are sorry for the confusing description. The representative pictures for the follicles at different stages were provided in Author response image 1. The oocytes in ovaries were labeled with MVH antibody. B, C, and D are the magnified views of the box regions in panel A. Primordial follicles were defined as an oocyte surrounded by a layer of squamous (flattened) granulosa cells as shown in panel B and D (arrowheads). Primary follicles possessed an oocyte surrounded by a single layer of cuboidal granulosa cells as shown in panel B (arrows). Secondary follicles were surrounded by more than one layer of cuboidal granulosa cells, with no visible antrum as shown in panel D (arrows). Antral follicles contain multiple layers of granulosa cells and visible antral spaces as shown in panel C (arrows). The description of “preantral follicle” is confusing and it has been removed in the revised manuscript.

**Author response image 1. sa2fig1:** 

(2). Provide a quantification of follicles at different stage presented in control and mutant animals at different times (2 weeks, 3 weeks, …) in Figure 1.

Thank you very much for the suggestion. The quantification of primordial, primary, secondary and antral follicles in control and *Prmt5^flox/flox^;Sf1-cre* female mice at 2, 3, 4 and 5 weeks of age has been added to the revised manuscript as Figure 1—figure supplement 4.

2. Line 149: Please provide more details about the fertility test. I wonder when the follicles were all lost in mutant mice. Figure S4 showed that the apoptosis and proliferation of granulosa cells was not changed in Prmt5flox/flox; Sf1-Cre mice at P14 and P18.

*Prmt5^flox/flox^;Sf1-Cre* female mice were infertile. As suggested by the reviewer, the results of fertility test have been added to the revised manuscript as Figure 1—figure supplement 3.

In this study, we found the apoptosis and proliferation of *Prmt5*-deficient granulosa cells were not changed at P14 and P18. But the identity of *Prmt5*-deficient granulosa cells was changed at P18, which resulted that the secondary follicles could not progress to the antral stage (Figure 1, Figure 1—figure supplement 4). Apoptosis is probably the fate of the premature luteinized granulosa cells in the arrested follicles at later stage. At 4 weeks of age, although the number of secondary follicles is not decreased, most of them are abnormal with less layers of flattened granulosa cells. The number of secondary follicles evidently decreased at 5 weeks. But the primordial and primary follicles are always present in *Prmt5*-deficient ovaries.

3. Please specify clearly which band is WT1 and FOXL2 in Figure 5A.

For WT1 protein, two bands represent two isoforms of WT1 protein, the upper band is +exon5 (54KD), the lower band is -exon5 (52KD). For *FOXL2* protein, the two bands are very close. Both of them were detected for every replicates, and the expression changes were the same with different treatments. Therefore, we speculate they are most likely different forms of post-translational modification of *FOXL2* protein.

4. Please provide a little bit more background about the Sf1 Cre mice and point out when and where the Cre recombinase expressed.

Thank you very much for the suggestions. *Sf1-cre* mouse strain was generated by Parker lab (Genesis, 2006, doi:10.1002/dvg.20231). In this mouse model, *Cre* recombinase was first expressed in the adrenogonadal primordium at 10 dpc. In gonads, *Cre* recombinase was expressed in the somatic cells. Other sites of expression include anterior pituitary, the ventromedial hypothalamic nucleus, and the spleen. The background about *Sf1-cre* mouse was included in the introduction of revised manuscript.

We also analyzed PRMT5 expression in theca-interstitial cells of *Prmt5^flox/flox^;Sf1-cre* ovary. As shown in Author response image 2, in *Prmt5^flox/flox^;Sf1-cre* ovary, PRMT5 expression was lost in granulosa cells (Author response image 2, arrowhead, red), but PRMT5 expression in 3β-HSD-positive theca-interstitial cells was not affected (Author response image 2, arrows). To further exclude the possibility that the defect of follicle development in *Prmt5^flox/flox^;Sf1-Cre* mice was caused by inactivation of *Prmt5* in theca-interstitial cells, *Prmt5^flox/flox^* mice were crossed with *Cyp17a1-cre* mice (theca-interstitial cell specific *Cre* mice). We found that female *Prmt5^flox/flox^;Cyp17a1-cre* mice were fertile, the histology of the ovaries was normal (Author response image 3). These results indicated that the defect of follicle development in *Prmt5^flox/flox^;Sf1-cre* mice was caused by inactivation of *Prmt5* in granulosa cells.

**Author response image 3. sa2fig3:** 

5. Please provide more information about how quantification of protein levels on WBs is done.

Thank you very much for the suggestion. Equal amounts of total protein was used for the Western blot analysis. Blots are representative for three independent experiments. Densitometry was performed using ImageJ software. The protein level was normalized to GAPDH. The normalized level of one sample was set 1, the ratio of the normalized levels of other samples to that of this sample was the relative expression level. Unpaired two-tailed Student’s t-tests were used for comparison between two groups. For three or more groups, data were analyzed using one-way ANOVA.

6. For Figure 6D-F, the authors use IgG as negative control, and I think the Prmt5flox/flox; Sf1-Cre granulosa cells are a better control for the co-immunoprecipitation. In Figure 6F, the western blot results of HnRNPA1 and SYM10 are from the same membrane? I wonder what the expression of HnRNPA1 and SYM10 are in Prmt5-deficient granulosa cells.

Thank you very much for the suggestion. As suggested by the reviewer, we have repeated the co-immunoprecipitation experiments using *Prmt5^flox/flox^;Sf1-cre* granulosa cells as a negative control, and the results were added to the revised manuscript as Figure 5E.

In the former Figure 6F, the western blot results of HnRNPA1 and SYM10 were from different membranes. Because HnRNPA1 and SYM10 are at the same site on the membrane, Western blot analysis of these two protein was performed using different membranes with same amount of loading protein.

The total protein level of HnRNPA1 did not change in *Prmt5^flox/flox^;Sf1-cre* granulosa cells. But the symmetric dimethylated HnRNPA1 (SYM10) was decreased in *Prmt5^flox/flox^;Sf1-cre* granulosa cells. *Prmt5* loss can not influence HnRNPA1 protein expression, only affects its symmetric dimethylation state.

7. It is crucial to analyze the methylation status of HnRNPA1 in P14 and P18 Prmt5-deficient granulosa cells.

Thank you very much for the suggestion. As suggested by the reviewer, we have analyzed the methylation status of HnRNPA1 in *Prmt5*-deficient granulosa cells at P14 and P18. Co-immunoprecipitation experiments were performed with HnRNPA1 antibody and symmetric dimethylation of HnRNPA1 was analyzed with SYM10 antibody using Western blot analysis. As shown in Figure 5E of revised manuscript, the symmetric dimethylation of HnRNPA1 in *Prmt5^flox/flox^;Sf1-cre* granulosa cells was significantly decreased at P18.

The results in P14 *Prmt5*-deficient granulosa cells were shown in Author response image 4. The level of symmetric dimethylation of HnRNPA1 was also decreased at P14. Although the expression of WT1 protein was not significantly decreased in *Prmt5^flox/flox^;Sf1-cre* granulosa cells, the level of symmetric dimethylation of HnRNPA1 has already decreased at P14, indicating that the decrease of dimethylation of HnRNPA1 occurs early than decrease of WT1 expression.

**Author response image 4. sa2fig4:** 

8. Line 585, A "secong" enzyme?

We are sorry for the spelling mistake. We have corrected it as “second” in the revised manuscript.

9. Line 537, the authors mentioned that "the inhibition of HnRNPA1 ITAF activity by PRMT5 does not occur through changes in the binding of HnRNPA1 to Wt1 mRNA", please add some discussions about the possible regulation mechanism between the HnRNPA1 and WT1.

As suggested by the reviewer, we have modified the discussion in the revised manuscript.

Reviewer #3:In their manuscript entitled "PRMT5 regulates ovarian follicle development by facilitating Wt1Translation" Chen and colleagues test the hypothesis that granulosa cell-specific expression of PRMT5 is required for ovarian follicle development in the mouse. To answer this question, the authors successfully used a Sf1-CRE driver to selectively disrupt a floxxed allele of PRMT5 in granulosa cells. The manuscript is roughly divided into two main sections. The first section addresses ovarian follicle development defects associated with the granulosa cell-specific ablation of PRMT5 and the second section defines the molecular details associated with the gene expression changes.It appears that the authors have completed and presented a series of important experiments that are novel and of potential high impact. The manuscript is well written and clear and the data are of excellent quality. The study is unique in that it includes highly complicated set of integrated in vivo and in vitro studies to identify molecular mechanisms underlying the regulation of ovarian follicle development. The major findings and conclusions are largely supported by the data that is presented in the manuscript and these conclusions are highly impactful to the field of regulation of gene expression underlying mammalian reproduction and development.One important concern is that although many clear data are presented, the quantitative and reproducible nature of this data remains questionable. In many cases it is unclear in how many biological or experimental replicates the data is presented from and if sufficient replicates were used to make these important conclusions. This is true for the in vivo mouse data where clear examples of the effects are shown without the necessary quantification or sufficient replicate numbers to make the strong conclusions stated. This is especially true for assessing ovarian follicle numbers between individuals that do vary even within genotypes. In similar fashion, the manuscript hypothesizes the existence of a novel IRES in the 5'UTR of WT1 presents strong evidence for it but never attempts to identify this IRES. The absence of some minor controls and obvious tests of their hypothesis is also a bit concerning. For example, while the rescue experiments with the cDNA for WT1 are really exciting, but this construct missing the 5'UTR of WT1 should be insensitive to the PRMT5 inhibitor but this is not tested. The absence of these controls or additional experimental manipulations detracts from the potential high impact of these findings.While the granulosa cell targeting of PRMT5 looks successful in Figure S2, Figure 1 is highly descriptive in nature and not sufficiently quantitative. Numbers of each follicle type from multiple biological replicates and fertility status of multiple replicates of each genotype must be completed over a significant period of time. Mating data concluding infertility is nowhere to be found in the manuscript.

Thank you very much for the suggestion. The quantification of primordial, primary, secondary and antral follicles in control and *Prmt5^flox/flox^;Sf1-cre* female mice at 2, 3, 4 and 5 weeks of age has been added to the revised manuscript as Figure 1—figure supplement 4.

The fertility test of *Prmt5^flox/flox^;Sf1-cre* female mice has been performed before submitting our manuscript. *Prmt5^flox/flox^;Sf1-Cre* female mice were infertile, the results of fertility test have been added to the revised manuscript as Figure 1—figure supplement 3.

Reduction of WT1 in the PRMT5 KO at P18 is interesting, but how many replicates of each genotype were assessed and can the staining be analyzed in a more quantitative nature in Figure 2.

Three to five mice for each genotype were used for immunostaining and one representative image of similar results was presented. The results of immunohistochemistry are difficult to quantify. Therefore, we analyzed the mRNA and protein level of *Wt1* in granulosa cells by RT-qPCR and Western blot analysis. As shown in Figure 4 of revised manuscript, the protein level of *Wt1* was significantly decreased in *Prmt5*-defiicent granulosa cells, but the mRNA level of *Wt1* was not changed in *Prmt5^flox/flox^;Sf1-cre* granulosa cells.

Figure 4 seems redundant given the fact that follicle structures are disorganized in vivo unless a specific cellular process is examined. For example, does steroid biosynthesis increase in the cKO cultures vs. controls? There are a few key additions possibly already completed or that need to be completed that would enhance the strength of the stated developmental conclusions.

We agreed with the reviewer’s comments. This Figure has been moved to figure supplement as Figure 3—figure supplement 2.

We have tried to analyze the steroid biosynthesis in in vitro cultured follicles. However, the culture medium was very limited, it was not enough to do the analysis. Actually, we have analyzed estradiol level of ovary tissues at 3 weeks of age. As shown in Author response image 5, estradiol level was significantly increased in *Prmt5^flox/flox^;Sf1-cre* ovaries compared to the control. The data are presented as the mean ± SEM (n=5). *P<0.05.

**Author response image 5. sa2fig5:** 

Similar details of the molecular analyses are missing as well that would increase the strength of these conclusions. Is the inhibitor specific for PRMT5 or are other arginine methyltransferases inhibited that could affect the stated conclusions in the manuscript.

Thank you very much for the comments. EPZ015666 has been used in many studies to specifically inhibit PRMT5 (for example, Nucleic Acids Research, 2017, doi: 10.1093/nar/gkw1367; Nat Chem Biol, 2015, doi: 10.1038/nchembio.1810; Exp Eye Res, 2021, doi: 10.1016/j.exer.2020.108286; J Neurooncol, 2019, doi: 10.1007/s11060-019-03274-0; Leukemia, 2018, doi:10.1038/leu.2017.334. etc.). The specificity of EPZ015666 for PRMT5 was first validated by Chan-penebre et al., (Nat Chem Biol, 2015, doi: 10.1038/nchembio.1810).

In our study, we also used *Prmt5^flox/flox^;Sf1-cre* granulosa cells to test the functions of *Prmt5* on *Wt1* IRES activity and we found *Wt1* IRES activity was significantly decreased in *Prmt5^flox/flox^;Sf1-cre* granulosa cells. This data has been added to the revised manuscript as Figure 5—figure supplement 1C. These results were consistent with the experiments using EPZ015666 on granulosa cells. Moreover, the results of EPZ015666 on granulosa cell specific and steroidogenic gene expression at mRNA and protein level were also consistent with those in *Prmt5^flox/flox^;Sf1-cre* granulosa cells as shown in Figure 4.

In figure 6C, the CCND1 IRES is also predicted to be affected by this inhibitor but this is not tested or shown.

As suggested by the reviewer, the effect of EPZ015666 on *Ccnd1* 5’UTR IRES activity was examined in primary granulosa cells and the data has been added to Figure 5D of the revised manuscript. *Ccnd1* IRES activity was significantly decreased in granulosa cells after EPZ015666 treatment.

What is the source of the input differences in figure 6D and E? In 6D there is relatively equivalent levels of all three proteins and in E HNRNPA1 is highly reduced and does this affect the outcome of the IP? How many times was the experiment shown in 6F completed and what is the potential variability observed between replicates?

The co-immunoprecipitation experiments were all repeated for three times. Every time, ovaries of more than ten female mice were used for granulosa cell isolation and then co-immunoprecipitation experiments. The similar results were obtained for different replicates. The difference between former Figure 6D and 6E is probably because the exposure time for MEP50 and PRMT5 in Figure 6E was a little longer when scanning the NC membranes than Figure 6D. But wherever in 6D or 6E, nearly no proteins could be immunoprecipitated with IgG. From the results, we can see among the same amount of immunoprecipitated proteins, the amount of MEP50 and PRMT5 immunoprecipitated with MEP50 antibody is more than that immunoprecipitated with HnRNPA1 antibody. HnRNPA1 antibody is also effective because it can immunoprecipitate large amount of itself. One possible reason is that HnRNPA1 is abundantly expressed in granulosa cells, whereas only a small proportion of HnRNPA1 interacts with PRMT5 and MEF50 proteins at the moment. Therefore, HNRNPA1 antibody can pulldown more *HNRNPA1* protein than PRMT5 protein.

The experiments shown in Figure 6F were also repeated for three times and the results were very similar among the replicates. There was no obvious difference in HnRNPA1 protein expression between control and *Prmt5^flox/flox^;Sf1-cre* granulosa cells, but the symmetric dimethylation of HnRNPA1 was significantly decreased in *Prmt5^flox/flox^;Sf1-cre* granulosa cells.

As suggested by reviewer 2, we used *Prmt5^flox/flox^;Sf1-cre* granulosa cells as a negative control in co-immunoprecipitation experiments. The former Figure 6D-E has been changed into Figure 5E in the revised manuscript.

In figure 7A, is there a significant difference in WT1 expression levels or is there a relative increase? Presenting and/or quantifying multiple replicates of each manipulation might help distinguish between these two possibilities.

Thank you very much for the suggestion. There is a significant difference in WT1 expression in the former Figure 7A. As suggested by the reviewer, the results of Western blot have been quantified and the result has been added to the revised manuscript as Figure 6—figure supplement 1

In figure 8, is the adenovirus expression of WT1 cDNA without 5'UTR insensitive to the PRMT5 inhibitor?

In *Wt1*-expressing adenoviruses, we used CMV promoter to control *Wt1* cDNA expression and no 5'UTR was contained. So the expression of *Wt1* cDNA was not affected by PRMT5.

In *Prmt5^flox/flox^;Sf1-cre* granulosa cells infected with *Wt1*-expressing adenovirus, WT1 expression is dramatically increased, indicating *Wt1* expression following CMV promoter is not affected by PRMT5. We also tested it in EPZ015666-treated granulosa cells. As shown in Author response image 6 Western blot analysis, WT1 expression is significantly increased in granulosa cells after infection with *Wt1*-expressing adenovirus compared with control adenovirus, no matter whether or not the granulosa cells were treated with EPZ015666.

**Author response image 6. sa2fig6:** 

We also performed another experiment to try to identify the IRES in *Wt1* 5’UTR. *Wt1* 5’UTR was divided into three fragments and respectively inserted into pRF construct (pRWT1F -268~ -158, pRWT1F -198~ -58, pRWT1F -105~ 1). These constructs were transfected into primary granulosa cells. We found the luciferase activity of pRWT1F -268~-158 and pRWT1F -105~1 was significantly increased compared to control pRF, but it was much lower than that of the full length 5’UTR, suggesting the full length of *Wt1* 5’UTR is required for maximal IRES activity. This result about the IRES activity of the three fragments of *Wt1* 5’UTR has been added to the revised manuscript as Figure 5C.